# PRIOR-BASED NOISY TEXT DATA FILTERING: FAST AND STRONG ALTERNATIVE FOR PERPLEXITY

**Yeongbin Seo**      **Gayoung Kim**      **Jaehyung Kim** [†]      **Jinyoung Yeo** [†]
Department of Artificial Intelligence
Yonsei University
{suhcrates,kykim,jaehyungk,jinyeo}@yonsei.ac.kr

## ABSTRACT

As large language models (LLMs) are pretrained on massive web corpora, careful selection of data becomes essential to ensure effective and efficient learning. While perplexity (PPL)-based filtering has demonstrated strong performance, it suffers from drawbacks: substantial time costs and inherent unreliability of the model when handling noisy or out-of-distribution samples. In this work, we propose a simple yet powerful alternative: a **prior-based data filtering** method that estimates token priors using corpus-level term frequency statistics, inspired by linguistic insights on word roles and lexical density. Our approach filters documents based on the mean and standard deviation of token priors, serving as a fast proxy to PPL while requiring no model inference. Despite its simplicity, the prior-based filter achieves the highest average performance across 20 downstream benchmarks, while reducing time cost by over **1000×** compared to PPL-based filtering. We further demonstrate its applicability to symbolic languages such as code and math, and its dynamic adaptability to multilingual corpora without supervision. The code is available online (https://github.com/ybseo-ac/prior_filter).

# 1 INTRODUCTION

Large Language Models (LLMs) have achieved impressive performance by training on massive datasets, with web text serving as a primary data source. As web content continues to grow indefinitely, it offers unlimited data for pretraining. However, two major challenges necessitate careful filtering steps: (1) Web data is so large that we need to choose efficiently to save computational resources, and (2) It contains a lot of noise, which can harm the model if not properly filtered.

To address this need, various data selection methods have been proposed. Early approaches relied on heuristic rules (Raffel et al., 2023; Bane et al., 2022), but more recent trends have shifted toward model-based techniques (Xie et al., 2023b; Marion et al., 2023). These methods typically involve training a reference model on a target dataset and using it to identify desirable data. The model may perform binary classification (Xie et al., 2023a) or compute similarity with the reference dataset (Xie et al., 2023b). Among these, using the perplexity (PPL) score from a reference model as a criterion of filtering is currently known to offer the best performance while maintaining a relatively simple implementation (Ankner et al., 2024). We provide a more detailed review of related work in §A.

However, PPL-based approaches come with the following inherent limitations. (1) *Time cost*: These methods require training a reference model, followed by inference of PPL over the whole corpus. Given that web-scale data can easily exceed trillions of documents and continues to grow, performing inference over the entire corpus becomes prohibitively expensive. (2) *Reliability*: LLMs often fail to accurately assess samples from distributions that is not seen while training, such as noisy data. As a result, generative perplexity may sometimes assign high scores to noisy or low-quality text (Holtzman et al., 2020; Welleck et al., 2019). This issue might become more pronounced when using smaller models to reduce inference costs, further undermining reliability.

---

[†]Co-corresponding authors

To address this limitation of the PPL-based approach, we introduce a **prior-based data filtering** method grounded in linguistic insights. Instead of computing the full conditional probability of each token in the data $p(x_i|x_{<i}) \propto p(x_{<i}|x_i)p(x_i)$ ($x_i$ is token of a data d), this method focuses solely on estimating the prior term $p(x_i)$ with statistical metric such as term-frequency. It is extremely simple and significantly faster (almost **0.1% time consumption** compared to PPL-based), while it achieves even better performance on downstream task benchmarks.

Interestingly, this method is inspired by traditional techniques used in deciphering ancient languages. The 8th-century linguist Al-Kindi first proposed that, in order to decipher an encrypted language, analyzing the frequency of its words provides a clue (Al-Kadit, 1992). If some word appears with the highest frequency across multiple documents, it is likely to correspond to a function word, such as "is" or "a" in English. This indicates that term-frequency itself is a one-dimensional representation for the role of a word: high frequency maps to function words while relatively low frequency maps to content words (e.g., "US", "president"). Combining with another linguistic observation that well-formed sentences within a language tend to exhibit a consistent level of lexical density (i.e., ratio between function and content words) (Johansson, 2008), we can determine outlier document simply by computing the mean and variance of its term frequencies: which we term **prior-based data filter**.

The prior-based filter exhibits intriguing and practical properties, which we demonstrate empirically. (1) The linguistic principles underlying the term-frequency hold not only for English but also for other natural languages (e.g., Chinese and French), even for symbolic languages (e.g., code, mathematics). (2) Only a small amount of Chinese text data mixed into an English corpus may be noise and models can not learn patterns from it; however, as its amount increases, it becomes learnable by models. The prior-based filter is capable of automatically capturing this transition of learnability.

We demonstrate that models pretrained using the prior-based filter outperform models using the PPL-based filter, across 20 diverse downstream task benchmarks. Moreover, since token priors can be estimated from a relatively small corpus, the prior-based filter is approximately 1000 times faster, requiring only 0.25 hours compared to 216 GPU hours for PPL-based filtering on a 6B-token corpus.

Our contributions are as follows:

- We propose the prior-based filter as an approximate alternative to the PPL-based filter.
- We analyze the useful properties of the prior-based filter, including its efficiency and generalizability.
- Through extensive downstream benchmarks, we demonstrate that the prior-based filter is not only faster, but also outperforms the current state-of-the-art PPL-based filtering.

## 2 PRIOR IS A ONE-DIMENSIONAL REPRESENTATION FOR THE ROLE OF TOKEN

In this section, we first introduce PPL-based approach, which is the previous SOTA for data filtering. Then we define how to estimate the prior, a key component of PPL. We then analyze the linguistic properties and significance of the prior, to show its potential as an effective criterion for data filtering.

### 2.1 PPL-BASED APPROACH AND ESTIMATION OF PRIOR

The PPL-based filtering method is known as the most effective approach for filtering noise data from web text corpus for pretraining LLMs (Ankner et al., 2024; Marion et al., 2023). For the filtering, first, a small reference model $\theta$ (an autoregressive transformer architecture of 137M parameters) is trained on the corpus $D$. The model then computes the PPL for each data point $\mathtt{d} = (x_1, x_2, \ldots, x_N)$, where $x_i$ is the token at the $i^{th}$ position of a document, and $\mathtt{d} \in D$. Then, d with PPL values farthest from the median are discarded. Here, the PPL is defined as follows:

$$\mathrm{PPL}(\mathtt{d}) = \left[ \prod_{i=1}^{N} p_\theta(x_i|x_{<i}) \right]^{\frac{1}{N}} \tag{1}$$

$p_\theta(x_i \mid x_{<i})$ is the conditional probability of token $x_i$ given its preceding context $x_{<i}$ under the model $\theta$, that can be decomposed into likelihood and prior as follows.

$$p_\theta(x_i \mid x_{<i}) \propto p_\theta(x_{<i} \mid x_i) \cdot p_\theta(x_i) \tag{2}$$

In this Bayesian formulation, the likelihood term $p_\theta(x_{<i} \mid x_i)$ captures the dependency between the token $x_i$ and its preceding context $x_{<i}$, indicating how well the token aligns with the surrounding text. In contrast, the **prior** term $p_\theta(x_i)$ represents the marginal probability of the token $x_i$, independent of its context.

**Estimation of prior** Due to the independent property of prior, it is no longer necessary for a transformer model to learn the joint probability in order to estimate the prior. Therefore, in this work, we assume the prior $p_\theta(x)$ of a token $x$ is approximated by simple statistics (i.e., term-frequency) in a corpus $D$, estimated as follows: $p_{\text{prior}}(x) = \frac{f_D(x)}{\sum_{x' \in V} f_D(x')}$. Here, $f_D(x)$ is the number of occurrences of token $x$ in corpus $D$, $V$ is the vocabulary set.

## 2.2 FREQUENCY ANALYSIS IN LINGUISTICS

To justify the use of a token prior as a filtering criterion, we draw on linguistic insights that reveal its strong connection to lexical and syntactic structure. Linguistics offers two key insights related to term frequency, and by combining them, we can derive its potential utility as a data filtering criterion.

**(1) Term frequency is a 1-dimensional representation of a word's role:** The 8th-century linguist Al-Kindi first proposed an idea that is still widely used today (Al-Kadit, 1992): to decipher ancient or encrypted languages, analyzing the frequency of its words gives a clue. If some word appears with the highest frequency across multiple documents, it is likely to correspond to a **function word** (e.g., "is" or "a" in English) that serves grammatical roles. In contrast, **content words** which carry semantic meaning (e.g., "US", "president") tend to appear with relatively lower frequency. Therefore, frequency itself can serve as a basis for distinguishing between function words and content words. In other words, term frequency (i.e., prior) can be seen as a one-dimensional representation of a word's functional role. We analyze that this property partially stems from the next property.

**(2) Well-formed sentences typically exhibit a consistent range of lexical density:** As **lexical density** is defined as the proportion of content words against function words, it is known that well-formed sentences in a language typically maintain a certain range of lexical density (Johansson, 2008). From this, we can infer that broken and ill-formed sentences will deviate significantly from this range to be outliers.

By combining these two properties, we can derive a principle for identifying ill-formed documents. First, we use the token prior as a one-dimensional representation to estimate whether each token functions more like a content or function word. Then, by assessing the overall composition of function and content words, we can determine whether the document is an outlier.

## 3 PRIOR-BASED DATA FILTERING

In this section, we present an explanation and analysis of the prior-based data filtering method. (1) We first analyze the token-level term frequencies, demonstrating that linguistic insights are applicable at the token level. (2) We then apply this principle to build our filtering method. (3) Lastly, we validate its feasibility by analyzing data samples filtered by our approach.

In §C, we further provide a theoretical and empirical explanation that the prior-based filter generalizes across diverse types of languages and tokenizers.

## 3.1 ANALYSIS ON THE TOKEN PRIOR

We first analyze the token-level term frequencies by calculating the token priors (with formulation in §2.1) on the Dolma dataset (Soldaini et al., 2024). As we sort them by the logarithm (Figure 1), we can observe that the token priors distinctly fall into three clusters based on their height and slope, supporting the thesis that the token prior serves as a 1-dimensional representation for the token's role.

The three clusters in Figure 1 are as follows. (1) *High-prior zone*: a steep slope of high-prior tokens. We can observe that this zone mainly consists of function words (e.g., "the", "a", "is", "you"). (2) *Middle-prior zone*: As the priors in this zone have a similar range, they form a wide and gentle slope. This zone seems to mainly contain tokens for content words (e.g., "Phone", "shortcut", "tackles", "doorstep"). (3) *Low-prior zone*: The frequency is extremely low, and the slope becomes steep again. This region is primarily composed of accidental noise tokens (e.g., "=≛", "ÃĥÃĤÃĥÃĤÃĥ",

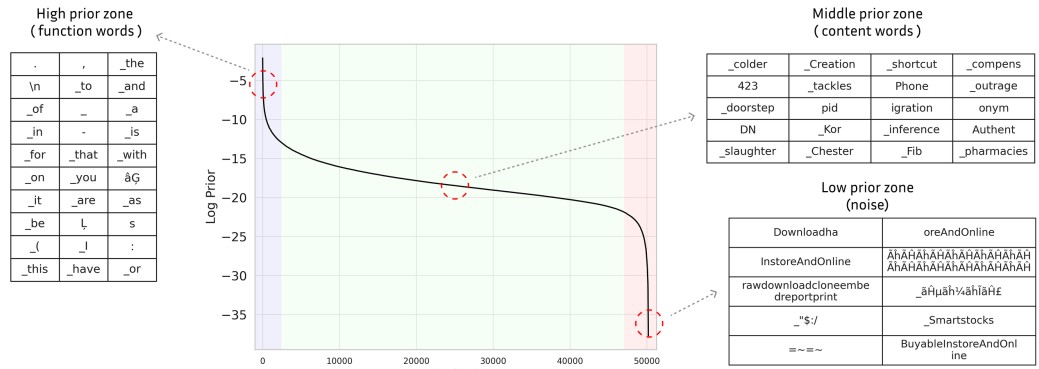

Figure 1: The line graph shows the logarithm of token priors (based on the GPT-2 tokenizer) computed from the Dolma dataset, sorted in descending order. The boxed regions highlight tokens from the top, middle, and bottom segments of the rank.

"ĠãĤ̂µãĥ̂¼ã"), including tokens from other language types that appear only a few times in the data (e.g., Chinese in English corpus).

## 3.2 FORMULATION OF PRIOR-BASED DATA FILTERING

We established two premises in §2.2: (1) A token's prior serves as a representation of its functional role, distinguishing function words from content words. (2) In a given language, a well-formed document typically maintains an average level of lexical density. By assessing the overall composition of function and content words, we can determine whether the document is an outlier.

However, since the prior is a continuous value rather than a discrete class label, we cannot directly compute the lexical ratio to assess the composition. Instead, we propose two alternative indicators to approximate the composition: the mean and standard deviation of token priors within a document.

**(1) Prior mean:** Since well-formed documents are clustered around a certain range of lexical density, the mean of token priors within such documents should also cluster around a certain value.
**(2) Prior standard deviation:** Given that well-formed documents tend to exhibit a stable lexical density, the variance (or standard deviation) of token priors within a document should also cluster around a specific value. We denote these metrics as $\mu_\mathrm{d}$ and $\sigma_\mathrm{d}$ respectively, formulating as follows. Specifically, we define the prior mean with a logarithmic transformation, as it aligns with the prior term in the PPL formulation; this is discussed in more detail in §3.4.1:

$$\mu_\mathrm{d} = \mathbb{E}_{x_i \in \mathrm{d}}\left[\log p_{\mathrm{prior}}(x_i)\right], \quad \sigma_\mathrm{d} = \mathrm{std}_{x_i \in \mathrm{d}}\left[p_{\mathrm{prior}}(x_i)\right], \quad \mathrm{d} \in D \tag{3}$$

As we assume that both $\mu_\mathrm{d}$ and $\sigma_\mathrm{d}$ of a well-formed document are clustered around certain central value, we define this central value as the median over the corpus $D$: $M_\mu = \mathrm{median}_{\mathrm{d} \in D}(\mu_\mathrm{d})$, $M_\sigma = \mathrm{median}_{\mathrm{d} \in D}(\sigma_\mathrm{d})$. The distance from the median is then used as a measure of outlierness. $\delta_\mu(\mathrm{d}) = |\mu_\mathrm{d} - M_\mu|$, $\delta_\sigma(\mathrm{d}) = |\sigma_\mathrm{d} - M_\sigma|$. To perform filtering, we discard the samples with the large $\delta$. The discarded portion is defined as the filtered set $F_\mu, F_\sigma$.

We analyze that the two criteria capture different aspects of the data. While $\delta_\mu$ captures the composition of tokens in a document, reflecting whether the document predominantly consists of high or low prior tokens, $\delta_\sigma$ reflects the distributional structure among tokens, indicating how uniformly or diversely the token priors are spread. This difference is also observed in the outlier samples.

In §B, we provide a theoretical derivation of the noise-detection behavior induced by $\mu_\mathrm{d}$ and $\sigma_\mathrm{d}$, and show that it approximates that of a PPL-based filter.

## 3.3 OBSERVATION ON DISTRIBUTION AND OUTLIER SAMPLES OF $\mu_\mathrm{d}$ AND $\sigma_\mathrm{d}$

We check whether the values of $\mu_\mathrm{d}$ and $\sigma_\mathrm{d}$ are clustered around central points, as hypothesized. For this, we randomly sample 600K examples from the Dolma dataset and compute $\mu_\mathrm{d}$ and $\sigma_\mathrm{d}$ for each d. As shown in Figure 2, both values exhibit broad distributions centered around their respective

medians, with relatively small deviations. Notably, beyond a certain threshold, we observe sharp increases in deviation, forming clear outlier regions (highlighted by red dashed circles). Upon inspecting these outlier samples, we find that they primarily consist of noisy documents lacking meaningful information (boxes of Figure 2).

Figure 2: The line graph displays the values of $\mu_{\mathtt{d}}$ and $\sigma_{\mathtt{d}}$ computed from token priors in the Dolma dataset, sorted in descending order. Boxes are outlier samples from both distributions.

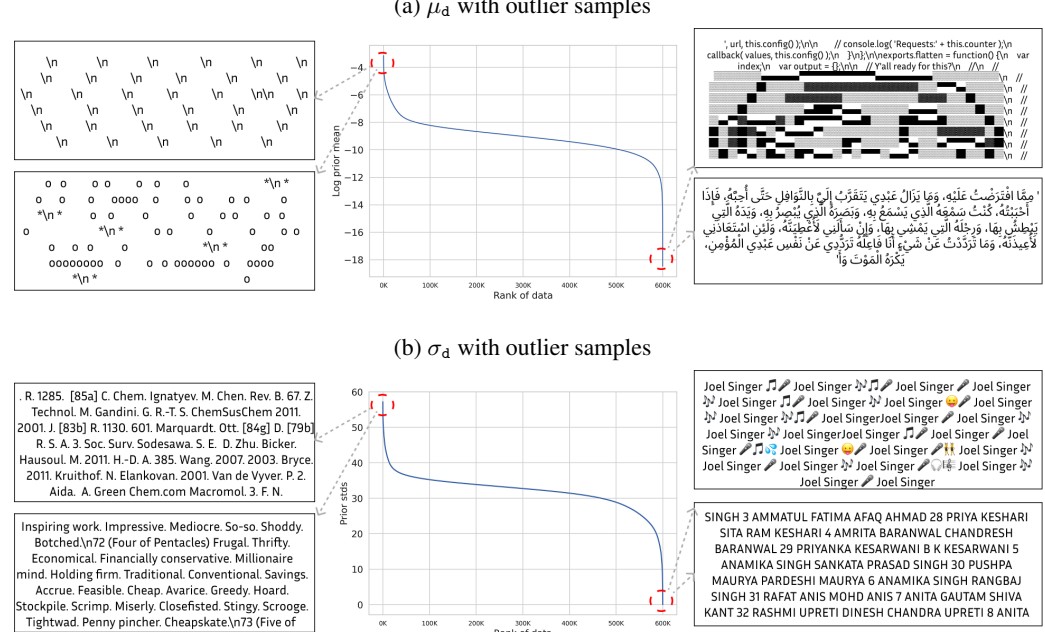

(a) $\mu_{\mathtt{d}}$ with outlier samples

(b) $\sigma_{\mathtt{d}}$ with outlier samples

**Characteristics of outliers from each metric**   We observe that the outliers for $\mu_{\mathtt{d}}$ and $\sigma_{\mathtt{d}}$ exhibit different characteristics. In the case $\mu_d$, the outliers tend to consist of tokens with either extremely high or extremely low prior values. For example, on the extreme-high side of the $\mu_d$ (left boxes of Figure 2a), documents mainly consist of line breaks ('\n') or space characters (' '), which is one of the tokens with the highest prior. On the extreme-low side (right boxes of Figure 2a), documents are often filled with non-English language or special characters.

Conversely, in the case of $\sigma_{\mathtt{d}}$, many outlier documents contain content-word tokens with middle-prior (boxes of Figure 2b). However, these words are arranged in unstructured ways, often appearing as a list of nouns without sentence structure. These differences arise because the $\mu_{\mathtt{d}}$ reflects the average composition of tokens in a document, whereas the $\sigma_{\mathtt{d}}$ captures the distributional pattern of those tokens. This suggests that both values should be used together for more effective data selection.

### 3.4 PROPERTIES OF PRIOR-BASED FILTER

#### 3.4.1 PRIOR-BASED FILTER APPROXIMATES PPL-BASED FILTER

The prior-based filter serves as an approximation to the PPL-based filter. We support this claim through both a formulation analysis and a statistical comparison of filtered data overlap.

$$\log \mathrm{PPL}(\mathtt{d}) \propto \underbrace{\sum_i^N \log p_\theta(x_{<i}|x_i)}_{\pi_{likelihood}} + \underbrace{\sum_i^N \log p_\theta(x_i)}_{\pi_{prior}} \tag{4}$$

First, the logarithmic form of PPL reveals that both the $\mu_{\mathtt{d}}$ and $\sigma_{\mathtt{d}}$ express two components of the PPL. (1) $\pi_{prior}$: The formulation of $\mu_{\mathtt{d}}$ in Equation 3 is exactly equivalent to the $\pi_{prior}$. (2) $\pi_{likelihood}$: as $\pi_{likelihood}$ captures the regularity of relationships among tokens within a document, $\sigma_{\mathtt{d}}$ similarly reflects the regularity in distribution of token priors. This suggests that the two measures are weakly aligned. Taken together, combining $\mu_{\mathtt{d}}$ and $\sigma_{\mathtt{d}}$ can serve as a reasonable proxy for perplexity.

**Prior can be even better metric than PPL**   While $\sigma_d$ captures an approximation of the likelihood term, it is significantly more saturated than the actual likelihood, which can be considered a limitation. However, conversely, the inherent instability of the $\pi_{likelihood}$ (described as follows) poses a limitation for the PPL-based approach. (1) When the model is small, it struggles to accurately learn the likelihood (Wei et al., 2022). (2) The model does not learn how to estimate likelihood for data from previously unseen distributions (mostly noisy data), which is not a problem for estimating only the prior. For this reason, previous studies have also reported that PPL often mistake repetitive or pattern-based noise as valid text (Holtzman et al., 2020). Empirically, the model trained with the prior-based filter shows better downstream performance than the one trained with the PPL-based filter (§4).

Figure 3: Extreme outlier samples selected based on three criteria, ensuring that each sample comes from a distinct criterion: PPL, $\mu_d$, and $\sigma_d$. ✓ indicates filtered out.

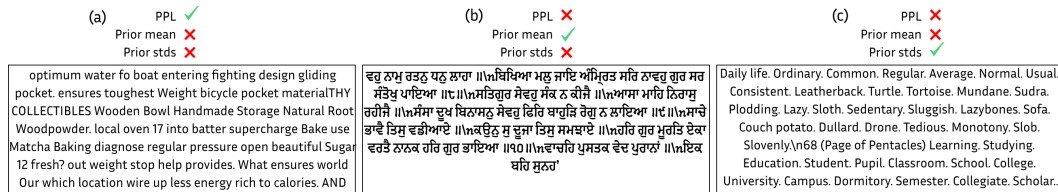

In §B, we provide a theoretical explanation of how the prior-based filter approximates the noise-detection behavior of the PPL-based filter, as well as the boundary conditions where a discrepancy between the two emerges. We also explain why median-based thresholding is robust.

**Observation on filtered samples**   This characteristic of PPL is also observed in outlier samples. We investigate the most extreme outlier samples from each metric (PPL, $\mu_d$, $\sigma_d$), excluding their overlaps (Figure 3). As described in §3.3, outliers of $\mu_d$ tend to be filled with extremely low or high prior tokens (Figure 3b), while those of $\sigma_d$ often consist of content words but lack function words or valid sentence structure (Figure 3c). In outliers of PPL (Figure 3a), content and function words appear to be well-balanced, giving the surface impression of well-formed sentences, but upon closer reading, many of them turn out to be semantically meaningless. This may reveal both a strength and a weakness of the PPL metric: it effectively captures subtle irregularities within well-formed documents, but may fail to detect noise arising from entirely out-of-distribution samples.

**Statistical comparison**   To demonstrate that prior-based filtering approximates the PPL-based filter, we measure the overlap ratio of data filtered by each metric. We first randomly sample 600K examples from the Dolma dataset. Then, for each value ($\mu_d$, $\sigma_d$, PPL), we extract the data points whose percentile rank falls within the top or bottom $\frac{e}{2}\%$ (Figure 4). These are denoted as the filtered sets $F_\mu$, $F_\sigma$, and $F_{ppl}$, respectively. For each filtered set, we compute the overlap ratio with $F_{ppl}$, defined as $\frac{|F \cap F_{ppl}|}{|F_{ppl}|}$.

The results show a strong correlation: When filtering by the $e = 0.10$, nearly 50% of $F_\mu$ and $F_{ppl}$ overlap. We also find $F_\mu$ aligns more closely with $F_{ppl}$ than $F_\sigma$. We provide additional evidence in §E.2

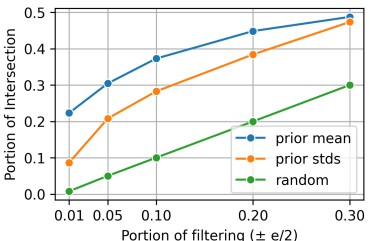

Figure 4: Overlap between outliers based on $\mu_d$ and $\sigma_d$ with those based on PPL, when filtering the top and bottom $\frac{e}{2}\%$ of samples (X-axis: $e$).

### 3.4.2   $\mu_d$ REFLECTS LEARNABILITY OF MINOR LANGUAGE IN MULTI-LINGUAL SETTING

The prior mean value has the property of dynamically reflecting the learnability of a data cluster (i.e., language type), especially when multiple clusters with distinct characteristics are mixed. For example, consider a corpus primarily composed of English data with a small portion of Chinese data included. While the Chinese samples may contain meaningful content, if their quantity is too small, the model will fail to learn the pattern of language. In this case, Chinese data is no more than noise. However, once the volume of Chinese data increases sufficiently, the model becomes able to interpret the language, making it learnable and meaningful data.

The prior-based filter captures this dynamic behavior without any special tuning. As shown in Figure 3, prior mean values tend to classify non-English samples as noise when they are sparsely mixed into English data. However, when the proportion of such data exceeds a certain threshold, the filter begins to treat them as valid language rather than noise.

To demonstrate this, we add a Chinese dataset (Wiki-ch)[1] to an English corpus (Dolma), with the Chinese data scaled to $a\%$ of the English corpus size. We then measure the percentage of added Chinese samples that fall into the outlier set (percentile rank falls within the top or bottom $10\%$). As shown in Figure 5, when the size of the Chinese data is only 1% relative to the English data, nearly all of it is classified as noise. However, once its proportion exceeds 20%, the rate of being classified as outliers drops to a level comparable to random filtering (10%, indicated by the red dashed line).

This characteristic offers a major advantage over methods that require manually specifying a reference dataset (e.g., DSIR (Xie et al., 2023b)). In DSIR, a human must decide whether to select English or Chinese data and then provide a suitable reference dataset accordingly. In contrast, the prior-based filter automatically determines whether a language should be filtered out based on its learnability.

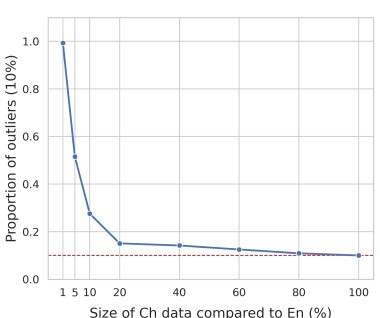

Figure 5: Proportion of Chinese data classified as outliers (Y-axis), after mixing Chinese and English data at a ratio of $a : 100$ ($a$ as X-axis). Outliers are the top and bottom 5% of $\mu_{\rm d}$.

### 3.4.3 FAST, SCALABLE FILTERING USING SUBSAMPLED PRIORS

One of the key advantages of the prior-based filter over model-based methods lies in its efficiency. Given the massive volume of new web data, which rapidly grows daily, training and inferring PPL value with a reference model can significantly amplify the time cost of filtering. In contrast, the prior-based filter only requires computing term frequencies and then calculating the mean and standard deviation of the priors.

Remarkably, the already minimal computation time of the prior-based filter can be further reduced. For a 6B-token corpus, the entire process takes about 35 minutes on 40 CPUs (Intel Xeon Silver 4210R @ 2.40GHz), which consists of two stages: assessing the token prior, and computing $\mu_{\rm d}$ and $\sigma_{\rm d}$. Among these, the most time-consuming step is the token prior assessing phase, which alone takes around 30 minutes.

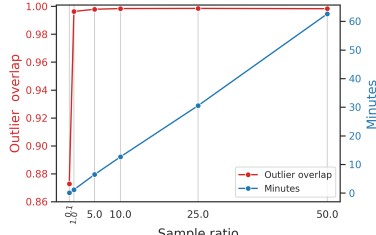

Figure 6: When token prior is computed with $b\%$ subset of Dolma (X-axis is $b$), the proportion of outliers overlapping with those from $b = 100$ is on the left Y-axis. The right Y-axis shows the computation time (in minutes) required to calculate the token prior at each $b$.

This assessment time can be significantly reduced, as term-frequency estimates remain highly consistent even when calculated from a small subset of the data. To verify this, we sample $b\%$ of a 6B-token dataset to compute the token prior and then measure how much the resulting outlier set (top/bottom 10%) overlaps with the outlier set derived from the full corpus ($b = 100$). As shown in Figure 6, even with just $b = 1\%$, the extracted outliers are nearly identical to those from full corpus; requiring only about 70 seconds, or roughly one minute.

## 4 EXPERIMENT ON DOWNSTREAM TASK

In this section, we evaluate the downstream task performance of models pretrained with different data filtering methods. Most training settings and hyperparameters follow those of "Perplexed by perplexity: Perplexity-based data pruning with small reference model (Ankner et al., 2024)". We first conduct experiments on a natural language

Table 1: Dolma v1.6 composition and its proportions based on token count.

| Source | Document type | portion |
|---|---|---|
| Common Crawl | web pages | 74.6% |
| The Stack | code | 13.4% |
| C4 | web page | 6.5% |
| Reddit | social media | 2.9% |
| PeS2o | educational papers | 2.3% |
| Project Gutenberg | books | 0.2% |
| Wikipedia, Wikibooks | encyclopedic | 0.1% |

---

[1] https://www.kaggle.com/datasets/notoookay/chinese-wikipedia-2023

(specified to English) web corpus, Dolma (Soldaini et al., 2024). This allows us to assess the effects on general language capabilities of a model (e.g., knowledge, language understanding, and symbolic understanding). To demonstrate that our prior-based method is applicable even to symbolic languages such as code and math, we also perform experiments on the Pile-github[2] dataset.

## 4.1 EXPERIMENT ON NATURAL LANGUAGE CORPUS AND GENERAL ABILITY

**Corpus setup** Following Ankner et al. (2024), we mainly use Dolma (Soldaini et al., 2024) as a pretraining corpus for a testbed of filtering methods. Dolma is a large-scale, diverse web-text corpus, designed for training and evaluating LLMs. It contains noisy web data sources that support general language use ability, such as world knowledge, commonsense reasoning, and symbolic problem solving. This corpus is composed of multiple web-scale datasets, including Common Crawl, Reddit, Wikipedia, and Wikibooks[3], The Stack (Kocetkov et al., 2022), C4 (Raffel et al., 2023), PeS2o (Soldaini & Lo, 2023), Project Gutenberg (Angelescu, 2013) (see Table 1). Among these, Common Crawl accounts for the major portion (74.5%) of the corpus. This makes it a particularly suitable environment for evaluating filtering methods, as it contains a high proportion of noisy web content that must be thoroughly filtered, while a small but valuable subset (e.g., books and educational data) must be preserved. For testing under resource constraints, we select v1.6—a smaller subset with 6.3B tokens. We divide this into blocks (d) of 512 tokens, and select a subset of $N$ tokens for pretraining.

**Baseline setup** When selecting a subset from Dolma, we follow the procedure defined by each method: (1) *no-filter*: Randomly selects $N$ without applying any filtering method. (2) *PPL-based*: Following the approach of Ankner et al. (2024) and §2.1, we first train a reference model (137M) on the random 3B tokens subset of dataset. We then compute the PPL score for each sample in the dataset. To obtain a final subset of size $N$, we discard samples with the highest and lowest PPL scores. (3) *DSIR*: Adopting the well-known method DSIR (Xie et al., 2023b), we estimate n-gram frequency from the reference dataset (we choose Bookcorpus and Wiki-en) and compute importance weights. DSIR is proposed as an advancement of FastText, a classifier-based method using manual data curation. (4) *prior-based* (ours): As described in §3.2, we first estimate token priors using a 10% subset of the full corpus. Based on these priors, we compute $\mu_d$ and $\sigma_d$ ($d \in D$). We then discard samples with the highest $\delta_\mu$ and $\delta_\sigma$ values in the constraint of $|F_\mu| = |F_\sigma|$, until the volume of final subset $|F_\mu \cup F_\sigma|$ reaches $N$. For all baselines, N is set to 50% (3B), which is observed in Ankner et al. (2024) to yield the best performance.

We use the GPT-2 architecture for pretraining, with large (1.5B) and small (137M) size models, using 8 GPUs (RTX A5000). Following Ankner et al. (2024), we set a max token length of 512, a global batch size 256, and a learning rate 2e-4, and train for 40K global steps (about 6B token duration). According to Ankner et al. (2024), the relative performance trends observed at 40K steps are maintained in later training steps.

As our study requires a significant amount of resources for pre-training across multiple baselines, we carefully adopted settings and observations from recent representative works to ensure experimental rigor under limited resources. We explain how our baseline lineup comprehensively covers the major streams of previous works in §A.2. We also provide justification of our experimental scale (e.g., training token duration, model size) in §F.2.

**Benchmark and evaluation setup** The types and settings of downstream tasks follow those used in the Ankner et al. (2024), based on the MosaicML evaluation gauntlet (MosaicML, 2023). Gauntlet includes tasks designed to assess five core capabilities: world knowledge, common sense reasoning, language understanding, symbolic problem solving, and reading comprehension. We normalize the accuracy of the individual task as $a_n = \frac{a_m - a_r}{1 - a_r}$, where $a_m$ is the accuracy of the model and $a_r$ is the expected accuracy of random guessing. We report the average normalized accuracy for each task, task category, and the average across all categories. Since some tasks are not proper for 1.5B models, we exclude benchmarks with average $a_n$ of baselines under 0.001. This results in a total of 20 benchmarks (details in §G)

**Results** As described in Table 2, the results show that the model trained with *prior-based* filtering achieves the highest average performance, with extremely small time cost. Key observations are as follows: (1) *DSIR* outperforms *no-filter*, and *PPL-based* outperforms *DSIR*, which aligns with

---

[2]https://www.kaggle.com/datasets/dschettler8845/the-pile-github-files-part-01
[3]https://commoncrawl.org/, https://www.reddit.com/, https://dumps.wikimedia.org/

findings from previous research (Ankner et al., 2024; Xie et al., 2023b). (2) *Prior-based* filter approximates *PPL-based* filter in principle, but yields better downstream performance. We analyze that this is because PPL score depends on the model's likelihood, which can be unstable. On the other hand, the prior is based on simple word frequencies, so it gives a more stable and reliable signal. (3) Though the *prior-based* model outperforms the *PPL-based* model in downstream performance, the *prior-based* filtering requires significantly less processing time. *PPL-based* filtering takes 216 GPU hours to select a 3B token subset ($20 \times 8$ GPU hours of training the reference model, $7 \times 8$ GPU hours of PPL inference), while *prior-based* filtering takes only 15 minutes (6 minutes of assessing token prior, 6 minutes of calculating $\mu_d$ and $\sigma_d$ in $D$)—under 0.1% of the time spent for PPL. This demonstrates the superior scalability and efficiency of our approach.

(4) In symbolic problem solving, *PPL-based* filtering performs the worst, whereas *prior-based* filtering performs competitively with other baselines. This suggests that PPL fails to capture small and meaningful segments of different types of data, while *prior-based* filtering is more robust in preserving them. This is due to the property of $\mu_d$ that reflects the learnability of multiple language types (§3.4.2). (5) While *no-filter* performs poorly across most abilities, it shows the highest score in symbolic problem solving. This might be because small but meaningful portions of data (e.g., math or programming-related) are par-

Table 2: Performance and time cost (for filtering) of the baselines pre-trained on Dolma across 20 benchmarks. The average normalized accuracy is the average of all categories. Avg: Avg normalized accuracy, W: World knowledge, C: Commonsense reasoning, L: Language understanding, S: Symbolic problem solving, R: Reading comprehension

| | Type | Time | Avg | W | C | L | S | R |
|---|---|---|---|---|---|---|---|---|
| **1.5B model** | | | | | | | | |
| vanilla | rule-based | - | 5.78 | 5.52 | 0.44 | 6.14 | **13.22** | 3.59 |
| FastText | classifier | 3.6 hrs | 7.09 | 6.71 | 6.11 | 6.89 | 11.93 | 3.82 |
| DSIR | n-gram | 4 hrs | 7.56 | 7.03 | 6.84 | 7.31 | 12.67 | **3.97** |
| PPL-based | model-based | 216 GPU hrs | 8.22 | **9.98** | **11.91** | 7.34 | 7.91 | 3.96 |
| Prior-based | (ours) | **0.25 hrs** | **9.20** | 9.53 | 11.27 | **10.31** | 11.13 | 3.79 |
| **137M model** | | | | | | | | |
| vanilla | rule-based | - | 4.96 | 4.96 | 1.81 | 1.47 | **12.83** | 3.70 |
| FastText | classifier | 3.6 hrs | 5.39 | 5.12 | 4.29 | 1.74 | 12.31 | 3.49 |
| DSIR | n-gram | 4 hrs | 5.60 | **5.68** | 4.93 | 1.97 | 11.60 | **3.80** |
| PPL-based | model-based | 216 GPU hours | 5.26 | 5.47 | 6.53 | 2.90 | 7.84 | 3.58 |
| Prior-based | (ours) | **0.25 hours** | **6.65** | 5.03 | **9.13** | **4.22** | 11.21 | 3.66 |

tially filtered out in other methods, but retained in the *no-filter*. For a *prior-based* filter, this issue can be handled by augmenting the small subset of the corpus for the targeted data type (i.e., datasets focused on coding or mathematics). This adjustment is straightforward and incurs minimal effort. (6) Across other skill categories, the *prior-based* method consistently outperforms other baselines or performs comparably to the best-performing one, resulting in the highest overall performance. (7) This trend remains consistent even for different-sized models.

**Further analysis** As data filtering removes noisy data, which is often found in outliers, it inherently risks discarding valuable minority data (e.g., a highly technical paper with rare terminology, a corpus including minority languages) as a trade-off. In §D, we analyze the robustness of the prior-based filter with respect to this trade-off.

In §C, we further provide a theoretical and empirical explanation that the prior-based filter generalizes across diverse types of languages and tokenizers. We also provide analysis on the sensitivity to factors such as block size and threshold in §E.

### 4.1.1 CONSISTENCY IN A LARGE-SCALE SETTING

We conduct experiments on larger models and datasets. For the efficiency of massive experiments, we adopt the strongest baselines from previous experiments (PPL-based and prior-based). We evaluate 3B (Qwen2.5-3B) and 1.5B models with training up to 12B token durations ($2\times$ repetitions). The results consistently demonstrate the superiority of the prior-based filter (Figure 7).

### 4.2 EXPERIMENT ON SYMBOLIC LANGUAGE CORPUS

We retain most of the settings from experiments of §4.1, including baselines and training configurations, but change the pretraining corpus to Pile-github. From the subset of 6B tokens, we extract a subset of 3B tokens with each filtering method. We exclude DSIR due to the difficulty of determining an appropriate reference dataset for Pile-github. This is also a critical limitation of the *DSIR*.

Pile-github mainly consists of code scripts, additionally containing a little mathematical data and natural language data. As it contains little information related to general language skills, such as world knowledge, we limit the evaluation only to 6 symbolic problem-solving benchmarks in gauntlet.

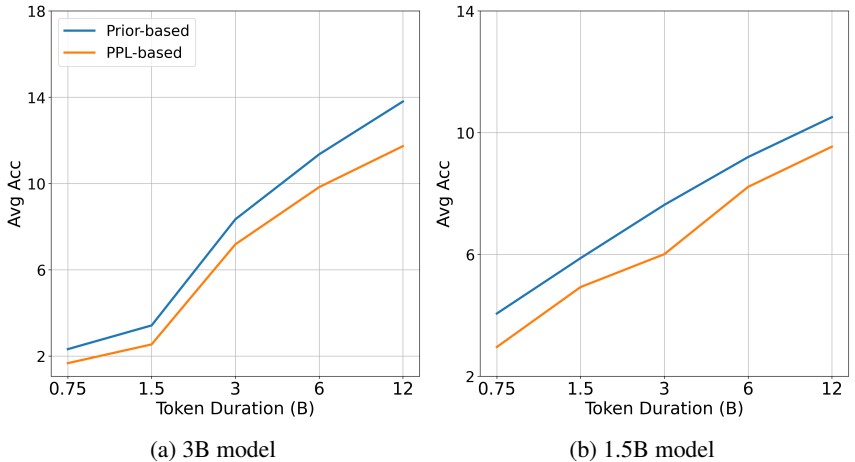

(a) 3B model        (b) 1.5B model

Figure 7: Average normalized accuracy across 20 benchmarks.

**Results** The observed results are as follows: (1) Consistent with the previous experiments, the *prior-based* method achieves the best performance with significantly less time than the *PPL-based* approach. (2) These findings suggest that our methods hold not only for natural languages (e.g., English, Chinese) but also for artificial symbolic languages (e.g., code, math). This means that well-formed data in a certain language type can be identified via *prior-based* statistics, regardless of language type. (3) Math-related benchmarks (BIG-bench elementary math QA, GSM8K, SVAMP) exhibit near-random performance across all baselines, likely because the Pile-github dataset consists predominantly of code scripts.

Table 3: Performance of the baselines pre-trained on Pile-github across 6 symbolic problem-solving benchmarks. CS: BIG-bench cs algorithms, Dyck: BIG-bench dyck languages, Op: BIG-bench operators, Ma: BIG-bench elementary math QA, GSM: GSM8k, SVA: SVAMP

|  | Time | Avg | CS | Dyck | Op | Ma | GSM | SVA |
|---|---|---|---|---|---|---|---|---|
| **1.5B model** | | | | | | | | |
| no-filter | - | 9.51 | 35.75 | 12.30 | 5.71 | 1.15 | **0.15** | **2.00** |
| PPL-based | 224 GPU hours | 11.21 | 37.42 | 20.60 | 7.14 | **2.09** | 0.00 | 0.00 |
| Prior-based (ours) | **0.26 hours** | **12.03** | **38.86** | 21.30 | **9.04** | 1.17 | **0.15** | 1.67 |
| **137M model** | | | | | | | | |
| no-filter | - | 10.15 | 37.87 | **16.30** | 5.23 | 1.52 | 0.00 | 0.00 |
| PPL-based | 224 GPU hours | 9.82 | **40.45** | 14.10 | 1.42 | 2.61 | **0.07** | 0.33 |
| Prior-based (ours) | **0.26 hours** | **12.19** | 40.22 | 16.00 | **7.14** | **3.08** | 0.00 | **6.66** |

## 5 CONCLUSION AND LIMITATION

We proposed a prior-based text data filtering method grounded in linguistic insight. The prior-based filter serves as an approximation of PPL-based methods, while achieving superior downstream performance and being over 1000× faster. Furthermore, it shows strong generalizability by performing effectively even on symbolic languages. This enables efficient filtering of rapidly growing web text data and provides a foundation for faster continual pretraining of LLMs.

However, since this method leverages linguistic properties, unlike other approaches such as PPL-based filtering or DSIR, it is less suited for extension to other modalities such as image data.

## ACKNOWLEDGEMENT

This research was supported by Institute of Information & communications Technology Planning & Evaluation (IITP) grant funded by the Korea government(MSIT), under AI Technology Development for Commonsense Extraction, Reasoning, and Inference from Heterogeneous Data(2022-0-00077, RS-2022-II220077, 50%), ITRC(Information Technology Research Center) support program(IITP-2025-RS-2024-00437102, 30%), Artificial Intelligence Graduate School Program (Yonsei University)(RS-2020-II201361, 20%). Jinyoung Yeo and Jaehyung Kim are the co-corresponding authors.

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

# A    RELATED WORKS

In this section, we first review two streams (e.g., rule-based and model-based) of previous works on web text data filtering for the pretraining of LLMs. Next, we explain that the baselines employed in our experimental setup cover all categories of existing baselines. Finally, we provide a more in-depth explanation of previous methods that share conceptual similarities with our proposed method.

## A.1    MAJOR STREAMS OF PRIOR WORK

**Rule-based**    Raw web-scraped data often contains a substantial amount of low-quality content, including documents with only space or machine-generated spam (Kreutzer et al., 2022). Because such noise can be detected with only very simple rules, the earliest filtering methods were rule-based. Raffel et al. (2020) introduced the following rules, which have subsequently been adopted in a similar form by most later works (Li et al., 2024a; Penedo et al., 2023; Chen et al., 2021; Rae et al., 2021; Bane et al., 2022): (1) Retain lines only with a terminal punctuation mark. (2) Discard pages with fewer than 3 sentences. (3) Keep lines with at least 5 words. (4) Remove pages with words from the bad-word list. (5) Removed lines containing the word "Javascript" and "lorem ipsum".

Because of its simplicity, such heuristic methods often fail to apply fine-grained filtering and risk discarding semantically valuable content inadvertently (Dodge et al., 2021). Nevertheless, because rule-based methods are extremely lightweight and require minimal computational resources, they are commonly applied at the web crawling stage (Bevendorff et al., 2018; Barbaresi, 2021). Consequently, most web datasets already have such methods applied by default (Penedo et al., 2023; Soldaini et al., 2024).

**Model-based**    More sophisticated approaches have been proposed that leverage the capabilities of deep neural networks (e.g., classifiers), achieving superior performance compared to heuristic filtering. Such approaches can be categorized into two groups based on whether they require a manually curated reference dataset.

(1) Without reference: A representative method is computing the perplexity of an LM on the text and filtering out outliers according to this measure. Another approach employs a linear classification method based on the embeddings of an LLM (Li et al., 2024a). AskLLM (Sachdeva et al., 2024) operates by presenting data points to the LLM and utilizing its reasoning capability to judge whether they constitute noisy data. EL2N (Paul et al., 2023) ranks samples based on the L2 distance between a model's prediction and the ground truth, thereby identifying data points that are more important for learning. Similarly, memorization-based methods (Biderman et al., 2023) assess how well a model memorizes token sequences within a document. PageRank score is a method of filtering documents based on how likely they are to be hyperlinked to other documents (Li et al., 2024a). Semantic Deduplication (semDedup) embeds each point in a dataset, clusters data points together, and removes data points within clusters that are too similar (Abbas et al., 2023).

(2) With reference: A well-known method is DSIR (Xie et al., 2023b), which constructs embeddings using either n-grams or FastText embeddings on a curated dataset, and evaluates the similarity of samples in the raw dataset to these reference embeddings. DSIR is proposed to advance FastText (Joulin et al., 2016); it trains the model to classify a curated dataset as either high quality or low quality.

Although model-based methods achieve high performance, they require training time and resources. In particular, when using LLMs for reasoning or embedding extraction, the process is more time-consuming, making it difficult to handle large datasets for pretraining. Moreover, when human curation is involved, performance becomes unstable and dependent on heuristic choices.

## A.2    EXPLANATION ON OUR BASELINE SETTINGS

As shown in Table 2, the baselines used in our main experiments include the vanilla Dolma dataset, FastText, DSIR, and PPL-based filtering. We explain how these methods comprehensively cover the major streams of baselines described above.

**(1) vanilla Dolma (noted as "no-filter" in the paper)**   Dolma (Soldaini et al., 2024) corpus itself already incorporates multiple filtering stages—including heuristic filtering, quality filtering, and even model-based filtering.

- Heuristic rule: adopted from rules used in C4, Gopher.
- Language filter (classifier-based) : Using fastText to select English data.
- Content filter (classifier-based) : Using fastText trained on the Jigsaw Toxic Comment dataset to filter out toxic documents
- Deduplication (rule-based) : duplication regarding an exact match of URL, document, or paragraph.

Our method achieves substantial performance gains on top of this. Moreover, these filters are usually computationally lightweight and are often applied already during the web-crawling stage (Li et al., 2024a; Soldaini et al., 2024). Therefore, they are compatible with our method by default.

**(2) DSIR**   DSIR operates on n-gram embedding vectors trained on a curated dataset. This work (Xie et al., 2023b) also compares a classifier-based embedding option (FastText). So this baseline covers both n-gram and classifier-based methods.

**(3) PPL-based**   The (Li et al., 2024a) paper compares a wide range of methods, and demonstrates that PPL-based approach outperforms well-known model-based methods (AskLLM, classifier on BGE features, PageRank, SemDedup) and heuristic filters, or is comparable to FastText and top-k average logits. Therefore, we selected the PPL-based method as the baseline representing other classifier-based methods.

**(4) FastText**   To directly compare against classifier-based approaches, we additionally provide the comparison with FastText (trained under the same dataset as DSIR). Its performance was similar but slightly lower than DSIR, which is consistent with the findings reported in (Xie et al., 2023b). The limitations of curation-based methods are also discussed in §A.3.3.

**Conclusion**   In summary, our experiments cover various types of baselines, including heuristic rule-based, n-gram-based, and classifier-based methods. In addition, we provide an extra classifier-based baseline experiment for the sake of clarity.

## A.3   METHODS WITH CONCEPTUAL RELATION

### A.3.1   DSIR

DSIR (Xie et al., 2023b) assumes that a well-curated reference dataset consisting of high-quality, well-formed text is available (Wikipedia and Bookcorpus is used in the original work). The method is to evaluate the similarity of sample $\mathtt{d}$ in the raw dataset to this reference corpus, and uses it as the filtering criterion.

According to Xie et al. (2023b), the process for estimating this similarity proceeds as follows. Given a corpus $D$, each document $\mathtt{d} \in D$ is sliced into a sequence of $n$-grams. For example, if the text input is "Alice is eating", it forms the list [Alice, is, eating, Alice is, is eating]. These $n$-grams are then mapped to hash indices, which are subsequently grouped into $m$ hash buckets (with $m = 10000$). The resulting hash frequencies form an $m$-dimensional categorical distribution vector $\gamma \in \mathbb{R}^m$, referred to as the feature distribution $P$. Separate feature distributions $P_{raw}$ and $P_{ref}$ are computed for the reference dataset and the raw dataset, respectively (each denoted as $q$ and $p$ in the original paper).

From the feature distributions, we can derive feature extractors $P(\mathtt{d})$as follows:

$$P(\mathtt{d}) = \prod_{j=1}^{m} \gamma[j]^{\mathtt{d}[j]} \tag{5}$$

$\mathtt{d}[j]$ indicates $j_{th}$ element of the sample $\mathtt{d}$. With this, we can calculate the importance weight for each data: $w(\mathtt{d}) = \frac{P_{ref}(\mathtt{d})}{P_{raw}(\mathtt{d})}$. The final selection is made by retaining those with the highest $w(\mathtt{d})$.

**Comparison with Our Method.** If we set the $n$-gram size to $n = 1$ and let the number of hash buckets $m$ equal the vocabulary size, the DSIR feature distribution $P$ essentially becomes the token prior used in our work. Moreover, the computation of our $\mu_{\mathsf{d}}$ (the mean log prior of tokens in a document) is conceptually similar to DSIR's feature extraction process.

However, our approach differs in several important ways: (1) Unlike DSIR, which requires both the feature distribution of the raw and the reference dataset, our method relies solely on the raw dataset. This reduces the dependency and effort for a high-quality refined reference. In practice, obtaining a truly noise-free dataset is difficult, as corpora like Wikipedia or BookCorpus (used in DSIR) also have noise. Furthermore, for diverse domains (e.g., GitHub, Chinese corpora), DSIR demands a separate domain-specific reference corpus, which introduces additional overhead and subjectivity in selecting appropriate reference data.

(2) DSIR typically uses bigrams ($n = 2$), while our method is based on unigrams ($n = 1$). As a result, function words in DSIR are often tied to neighboring content words and rarely appear independently in the feature distribution, like in the example [`Alice, is, eating, Alice is, is eating`]. Consequently, DSIR's distribution tends to reflect the frequency of content words while neglecting the function words. This indicates a difference in the filtering principle from our approach.

### A.3.2 SCDP

SCDP (Swift Cross-Data Pruning) (Nguyen & He, 2025) is a method that selects data based on the multivariate median of TF-IDF (term frequency and inverse document frequency) representations. This method selects data that is most similar to the dominant topic frequently covered in the corpus.

To describe the method, first, a feature vector $\mathbf{t}_i = TF_i \odot IDF_i$ is computed for each $\mathsf{d} \in D$. And documents that are closest to the median (multivariate median) are selected.

Compared to our approach, SCDP differs in a fundamental way: whereas we compute token priors based on $TF \odot DF$, SCDP uses $TF \odot IDF$, which is the inverse way of reflecting $DF$. Because tokens with high document frequency receive lower $IDF$ scores, the function words are downweighted or often entirely suppressed. As a result, SCDP's representation captures the frequency of content words only. This is in contrast to our method, which treats both function and content words as integral components of a document.

Such an approach leads to the following characteristics: (1) By eliminating the influence of function words, the method focuses on the composition of content words (i.e., topic), rather than on grammatical regularity. (2) Since selection is based on the median value, it favors documents that are closely related to one most frequent topic in the corpus.

This approach has a limitation in that the topic of the document does not necessarily correlate with its noise level. More specifically: (1) A corpus typically contains a diverse range of topics, some of which may be represented by only a small number of samples. If selection is based on topic similarity, informative but underrepresented data may be filtered out, even if it is not noisy. (2) Conversely, documents that align closely with the median topic can contain noise, while still being selected. For example, as exhibited in Figure 2b, certain web data consists of norm lists or repetitive content that may appear topically relevant but lack meaningful or well-structured information.

Due to these reasons, we argue that our approach is more optimal for identifying ill-formed, noise-heavy documents. This is because our method evaluates data based on whether the sentence is structurally well-formed, regardless of its topic.

### A.3.3 FASTTEXT

FastText (Joulin et al., 2016) is a model-based filtering method. To train the model, web-crawled data are assigned as "low-quality," while manually curated target data are assigned as "high-quality". And the classifier model is trained to infer the probability of a datapoint belonging to a high-quality set. (Xie et al., 2023b) explains that DSIR is an improved version of FastText, outperforming in performance. Like DSIR, a key limitation of FastText is that filtering quality is constrained by the accuracy of manual curation, which also increases rigidity and complexity due to the required human

labor. Moreover, because web-crawled data must be additionally tokenized, the computational time cost increases.

We provide additional analysis on the FastText model in the same setting as DCLM (Li et al., 2024a), assigning web-crawled data (RefinedWeb (Penedo et al., 2023), which is based on Common Crawl) as the "low-quality" set, while ELI5 and OH-2.5 as the "high-quality" set.

**Case analysis**   We analyzed the outlier cases from the FastText classifier and observed several unexpected patterns. Among the samples classified as having a 0% probability of belonging to the high-quality set, we observed a substantial number of well-formed texts—particularly those resembling news articles (Case 1). Conversely, many samples classified as having a 100% probability of being high-quality were clearly noisy or nonsensical scripts (Case 2). We double-checked our implementation, but confirmed that these results were solely due to the model's inference behavior.

Case1. Data assigned a 0% probability of being high-quality (i.e., 100% belonging to RefinedWeb) — ranking at the absolute bottom of the distribution.

```
Paula's Choice - Donating $50,000 to the COVID-19 Solidarity Re-
sponse Fund for World Health Organization.
Pyer Moss - Pyer Moss has set aside $10,000 to get supplies for med-
ical workers while also converting their NYC office into a donation
center to store the supplies.  Using local factories, Pyer Moss is
creating 1,000 mask covers to send directly to front line workers.
With the help of Jen Rubio,
```

Case2. Data assigned a 100% probability of being high-quality — ranking at the absolute top of the distribution (1st rank).

```
'0][1-9]\\|[1][0-2])([0-2][0-9]\\|[3][0-1])\\\\\\\\s\\\\\\\\s
    ?([0-1]?[0-9]\\|[2][0-3]):[0-5][0-9]:[0-5][0-9])` | | |\n| `
    ddMMyy HH:mm:ss` | `(([0-2][0-9]\\|[3][0-1])
    ([0][1-9]\\|[1][0-2])[0-9]{2}\\\\\\\\s\\\\\\\\s
    ?([0-1]?[0-9]\\|[2][0-3]):[0-5][0-9]:[0-5][0-9])` | | |\n| `MMM
    d HH:mm:ss` | `(Jan\\|Feb\\|Mar\\|Apr\\|May\\|Jun\\|Jul\\|Aug\\|
    Sep\\|Oct\\|Nov\\|Dec)\\\\\\\\s\\\\\\\\s?([0]?[1-
```

We hypothesize the following reasons for this unpreferred behavior:

**(1) RefinedWeb contains a substantial amount of well formed data.**   To avoid the cost of human labeling, DCLM chose to label RefinedWeb (Penedo et al., 2023) as the low-quality set, while labeling OH-2.5 and ELI5 as the high-quality set. During inference, the model predicts the probability (0 100%) that a given text belongs to the high-quality set. However, RefinedWeb also contains a considerable proportion of well-formed documents. As a result, many samples that FastText classifies with 0% probability of belonging to the high-quality set are in fact well-formed and informative texts (Case1), often resembling those found in RefinedWeb.

One possible explanation is that news article–style texts are prevalent in web-crawled sources (e.g., RefinedWeb), but largely absent from curated datasets like OH-2.5 and ELI5, which mainly contain question-answering format. As a result, the FastText model may have implicitly learned to classify the news article (or other non-QA-style) format as belonging to low-quality set, leading to systematic misclassification.

**(2) Limited discrimination capacity.**   Another possible explanation is the prevalence of code and math-related content in OH-2.5. Since such sources are relatively less common in RefinedWeb, the model may have overfit to these symbolic patterns during training. However, due to the limited capacity of the small FastText model, it is unable to capture deeper coherence within symbolic

language. As a result, it may incorrectly classify meaningless noise that superficially resembles symbolic content (as in Case 2) as belonging to the high-quality set with 100% probability.

These results underscore the weaknesses of model-based methods, supporting the robustness of prior-based approach.

# B    THEORETICAL DERIVATION AND BOUNDARY CONDITION ANALYSIS

In this section, we show that (1) under the linguistic observation presented in §2.2, the prior-based metric shares noise-distinguishing behavior with PPL, regardless of the scale of the model or corpus. In particular, (2) we demonstrate that the distance from the central point is a distinguisher of outliers. (3) We also explain the boundary conditions under which this property does not hold.

## B.1    THEORETICAL GUARANTEE OF NOISE DETECTION BEHAVIOR

### B.1.1    THE COMPONENTS OF PPL AND THE APPROXIMATION VIA PRIOR-BASED METRIC

As is well known, PPL consists of the following two components: the likelihood and the prior term.

$\log PPL(d) \propto \sum_i \log p(x_{<i}|x_i) + \sum_i \log p(x_i)$

And in §3.2, we introduced two prior-based metrics:

(1) $\mu_d \propto \sum_i \log p(x_i)$

(2) $\sigma_d = std[p(x_i)] \propto \sum_i [p(x_i) - e_d]^2, \ e_d = E[p(x_j)]$

**Prior term of PPL**    As can be seen, the $\mu_d$ is identical to the prior term of PPL. Therefore, the prior-based filter partially approximates PPL.

**Likelihood term of PPL**    The $\sigma_d$ and likelihood share a similarity in that they reflect relationships among tokens rather than averaging the independent properties of individual tokens. However, it is difficult to formally prove a universal connection.

However, under the following assumptions and scenarios, they exhibit similar behavior in filtering noisy data.

**Assumptions:**    We assume very simplified settings of linguistic observation in §2.2.

(1) Suppose tokens consist of only two categories, F and C, where F represents function words and C represents content words. (2) Within any well-formed document, these two groups must follow a fixed ratio $r^*$ (e.g., F:C=4:1).

(3) Actually, data $d$ has a fixed length (e.g., 5), and has an observed ratio $r$. And $r$ has only two possible values: $r^*$(well-formed) or $\hat{r}$ (noise).

Example of $r^*$ : FFFCF, FFCFF

Example of $\hat{r}$ : FFFFF, FFCCF

(4) If the data of $r^*$ takes the big proportion in the corpus, both the empirical average of ratio and the trained model $p$ converge near to $r^*$.

(1)–(3) are simplified versions of the linguistic observation in §2.2, and (4) represents the minimal condition required for our method to operate. (Experiment in §3.4.2 shows the minimal proportion is around 10

**Property of conditional likelihood**    In this simplified setting, conditional likelihood is determined solely by $r$ (e.g., $p(\text{else} \mid C) \ll p(\text{FFFF} \mid C) \approx 1$ ).

Therefore, the distribution of $l_d = \sum_i \log p(x_{\neq i}|x_i)$ places data with r* near to 1, while placing $\hat{r}$ near the left tail. And since $r^*$ dominates the corpus (assumption 4), the overall distribution of $l_d$ becomes right-skewed.

Therefore, in the distribution of $l_d$, data with ratio $\hat{r}$ can be identified through the left tails that deviate far from the representative value (e.g., median) of the distribution.

**Property of $\sigma_d$**  For the $\sigma_d$, the computation is straightforward. Given a fixed occurrence probabilities (e.g., $p(C) = 1/5$ and $p(F) = 4/5$), the $\sigma_d$ are distributed around specific value (e.g., $E[\sigma_d] = 0.26$) where the ratio of data is $r^*$. And the samples that deviate far from this point can be distinguished as the one with $\hat{r}$.

Therefore, both the likelihood term and the $\sigma_d$ exhibit similar behavior in identifying $\hat{r}$.

**Property of $\mu_d$**  Under the FC assumption, we further explain the role of the prior term of PPL ($\mu_d$) in distinguishing $\hat{r}$. Suppose that F and C have fixed occurrence probabilities (e.g., $p(F) = 4/5$, $p(C) = 1/5$) and appear within each document according to those probabilities. Then, we can observe that the average of their occurrence probabilities tends to concentrate around a specific value (e.g., $E(\mu_d) = 0.68$).

If the observed ratio deviates from this central value, it will move further away from the central value, allowing us to identify $\hat{r}$ accordingly.

**In conclusion, under assumptions 1–4, we can guarantee two properties of prior-based metrics.** (1) the prior-based metrics ($\mu_d, \sigma_d$) align with all components of PPL in identifying $\hat{r}$. So, we can say the prior-based filter "weakly aligns" with PPL-based filter. (2) the prior-based metric of normal data with valid structures ($r^*$) converges around a single central point, allowing outliers ($\hat{r}$) to be distinguished by their distance from this center.

### B.1.2  Discrepancy from theoretical guarantee: boundary condition analysis

However, such an approximation required simplified assumptions. Consequently, when these assumptions are violated in a real-world data, a discrepancy arises. The most significant violation case we can identify is the following: even when a document follows the ratio $r^*$, it may still contain no meaningful information. This corresponds to cases such as:

(1) Text with an appropriate balance ($r^*$) of function and content words, yet are semantically nonsensical. This case is illustrated with Figure 3(a). (2) The data are not noise, nor semantically nonsensical; rather, they are either too difficult or too trivial to be useful for the model. (In this scenario, the main LLM should compute PPL directly, so this case is not applicable to our PPL baseline either.)

In such cases, PPL will play a clearer role, and the prior-based filter will not be able to handle this, and we do not claim this. Prior filter is designed to identify only the noisy data ($\hat{r}$), as we emphasized in the title and §1.

However, we claim the following: (1) The prior can at least provide a clearer signal in identifying $\hat{r}$, as model-predicted likelihood can often be unreliable. (2) In the corpus with a very high proportion of noise (e.g., web crawled corpus), identifying $\hat{r}$ may be more impactful than deliberate reordering among $r^*$ data. (3) Still, compatible with PPL-based filter.

### B.2  Robustness justification for median-based thresholding

By drawing on the above theoretical arguments, we can justify the robustness of our methodological choices of a median-based threshold. For example, this allows us to answer the question of whether the median-based threshold remains effective when the corpus exhibits a mixed multimodal distribution.

### B.2.1  Mixed multimodal distributions

**(1)** Prior-based filter only measures the grammatical structure via the function-content word ratio. And texts in one specific language tend to share the same grammatical structure, regardless of their style or domain (e.g., news, forum). Thus, prior-filter will generalize to each other.

Dolma is such an example. Although Dolma consists of various types of data (e.g., news, books, symbolic scripts), our method has been shown to perform well across these domains.

**(2)** Std metric ($\sigma_d$) measures the distance between token priors, thus the distance itself may not vary much among other types of language.

Let's bring the "FC" assumption from §B.1 to here. Let's assume there are heterogeneous language groups A and B, and each has its function-content words pair, and each group follows the assumption of §B.1. $\{F, C\} \in A$, $\{R, T\} \in B$.

According to the linguistic axiom, we can assume that distinct linguistic system (A, B) shares a similar function-content ratio (e.g., 1:4). Therefore, the central value of std metric for both languages is the same (e.g., $E[\sigma^A] = E[\sigma^B] = 0.26$). Thus, $\sigma_d$ for A and B share the same central value, even though they are heterogeneous.

**(3)** However, in practice, if the amount of B is significantly smaller than A, then there can be an imbalance of the function-content ratio. For example, $p(F) = 0.4, p(C) = 0.1, p(R) = 0.0004, p(T) = 0.0001$. In this case, the central value of the distribution of $\sigma^A$ and $\sigma^B$ will diverge. We suggest two points:

**(3-1)** We tested an extreme case of this in §3.4.2 with the Chinese-English set, to answer the question "When two entirely different languages are mixed together, to what extent can a single median encompass both languages?" Our experiments indicate that if a minority group is mixed only over 10%, most of the minority language is still considered an inlier when using a single median.

**(3-2)** If this is still insufficient, we can compute the token prior by mixing in a corpus that shares attributes with the target corpus. For example, if we mix in more data that resembles corpus B and then recalculate the token prior, we might obtain token priors such as $p(F) = 0.4$, $p(C) = 0.1$, $p(R) = 0.4$, and $p(C) = 0.1$. In this case, we can bring corpora A and B to the same median value again, even though the actual proportion of B is very small.

This approach corresponds to the method described in §D.1(2) of our paper. For example, when we compute the token prior using a setting where GitHub data are added to Dolma, the resulting token prior enables better identification of GitHub-like data (e.g., symbolic scripts) within Dolma.

## C  GENERALIZATION TO OTHER LANGUAGES AND TOKENIZERS

In this section, we discuss how our method generalizes to languages beyond English and to tokenizers other than the GPT-2 tokenizer.

As we discussed in §B, our method is supposed to work with any language system following two features (1) the language is composed of function words and content words, (2) function and content words have distinct distribution of frequency. This character applies to most natural languages (e.g., Chinese, Japanese, Turkish).

**Most languages are composed of function-content words**  It is well observed in linguistics (Stromswold, 1994; Croft, 2002) that no matter how compound a language is, it can be separated into function morphemes and content morphemes.

For instance, the Turkish compounding word *evlerimizden* ('from our homes') decomposes into *ev* (content morpheme) + *ler* + *imiz* + *den* (grammatical morphemes).

These function and content words follow the frequency rule (i.e., function words tend to have higher frequency). For example, in Turkish, the function word "ler" appears across many different nouns (e.g., *evler*, *şehirler*, *öğrenciler* ), resulting in higher term frequency.

The same phenomenon occurs even when dealing with English. For example, a compound word such as *childish* is typically segmented into *child* and *-ish*. The suffix *-ish* appears across many different content words, and thus behaves similarly to a high-frequency function morpheme.

**The tokenizers split words into morphemes based on frequency**  Most tokenizers (e.g., BPE, WordPiece, SentencePiece) have tendency to merge characters with high co-occurrence probability into a single token (e.g., "i" + "n" → "in"). This is because most tokenizers, either directly or approximately, follow the Minimum Description Length (MDL) (Hou et al., 2023) objective, which aims to minimize both the vocabulary size and the length of representation. For example, the token

"*ler*" frequently co-occurs with various words, so assigning it as an independent token is more efficient (i.e., covering a wider range of expression with limited vocab) than assigning "*şehirler*" and "*evler*" as separate tokens.

Combining together, most of natural languages can be segmented into function and content components, and most tokenizers effectively reflect this distinction, even in morphologically compounding languages.

## C.1 EMPIRICAL EVIDENCE

**Other languages** The Chinese–English experiment (§3.4.2) provides evidence for generalization to other languages, such as Chinese.

Suppose that the prior-based filter works only for English but not for Chinese, then our filter will classify all Chinese data as noise. Then, in Figure 5, at the point where Chinese and English are mixed at a 10:100 ratio (at any ratio), nearly 100% of the Chinese data should be classified as outliers. However, the actual results show that most of the Chinese data were classified as inliers, only if this language group takes more than 10% of the corpus. Therefore, the prior-based filter works for Chinese as well.

Table 4: Same setting as Figure 5. Proportion of Turkish data classified as outliers after mixing Turkish and English data at a ratio of $a : 100$. Outliers correspond to the top and bottom 5% of $\mu_d$.

| Size of Turkish data ($a$) | Outliers (10%) |
|---|---|
| 1 | 0.5036 |
| 5 | 0.2778 |
| 10 | 0.2448 |
| 20 | 0.1984 |
| 100 | 0.1337 |

We also conducted the same experiment on Turkish (Table 4). And the results were consistent with those of the Chinese–English experiment. We provide an explanation below.

**Other tokenizers** We provide comparison with two additional tokenizers: LLaMA-3-8B (UTF-8-based, vocab size 128K), and T5-small (SentencePiece-based, vocab size 32K), in Table 5. GPT-2 tokenizer (mainly used in our paper, 50K) has a vocab size of 50K. All of these tokenizers tend to assign function morphemes as individual tokens in accordance with the MDL principle. However, we measure the performance differences among different granularities.

The result is reported in ETable 4 (also Table 4 of our paper). All tokenizers showed the advantage of filtering, while T5 exhibited slightly lower performance gain. We propose two possible interpretations: (1) There may exist a threshold of optimal granularity, which lies between 32K and 50K. (2) Using the tokenizer paired with the model may offer stability.

Nevertheless, as demonstrated, priors can be computed even with tokenizers mismatched to the model. If an optimal tokenizer exists, it can be freely adopted at any time.

Table 5: Performance of large (1.5B) models on Dolma under different tokenizers. Avg: Avg normalized accuracy, W: World knowledge, C: Commonsense reasoning, L: Language understanding, S: Symbolic problem solving, R: Reading comprehension

| Tokenizer | Avg | W | C | L | S | R |
|---|---|---|---|---|---|---|
| GPT2 | 9.20 | 9.53 | 11.27 | 10.31 | 11.13 | 3.79 |
| LLaMA-3 | 9.39 | 9.54 | 11.16 | 10.78 | 11.86 | 3.64 |
| T5-small | 8.11 | 8.59 | 7.43 | 7.95 | 12.32 | 4.22 |

## C.2 SYMBOLIC LANGUAGE

Symbolic languages (e.g., codes) are a more worrisome case, as they are more different from English, and it has not been verified whether linguistic observation of natural language applies to them. Therefore, we conducted experiments on this extreme case (§4.2), and empirically proved our method still generalizes to symbolic language as well. We can expect that it will generalize to other natural languages more similar to English.

## D PRESERVING MINORITY DATA

As data filtering removes noisy data, which is often found in outliers, it inherently risks discarding valuable minority data (e.g., a highly technical paper with rare terminology, a corpus including

minority languages) as a trade-off. Minimizing this trade-off is the general goal of research on filtering. In this section, we discuss our prior-based filtering method from the view of this trade-off.

**This trade-off can be measured through Dolma**    In our experiments on Dolma, we also observe such a trade-off. As shown in Table 1, Dolma consists primarily of English-based web data, with a small portion of programming language data (The Stack), and understanding this language is separately measured through "symbolic problem-solving" in Table 2.

**Prior-based filter shows a better trade-off**    Table 2 shows that the no-filter setting achieves the highest symbolic understanding, while the PPL-based filter shows very low symbolic understanding, even with the improved overall score. This clearly illustrates the trade-off: as filtering is applied, minority languages are pruned away. In contrast, the prior-based filter shows much higher symbolic performance compared to the PPL-based filter, while also maintaining a higher overall score. This indicates a significantly better trade-off of our method. We provide additional evidence in §D.2.

Nevertheless, we propose additional approaches to further address this trade-off.

## D.1    ADDITIONAL METHODS TO ADDRESS TRADE-OFF

**(1) Using only stds:**    Our original method leverages both the mean and standard deviations (stds) of the prior. Using only the stds may be beneficial, since the mean reflects the average frequency of tokens, whereas the std captures the dynamics among them. This distinction can make stds-based filtering more effective in identifying well-structured documents with low-frequency languages. In Table 6, the filter with stds as criterion achieves higher symbolic understanding than the mean+stds filter, while maintaining comparable average accuracy.

**(2) Calculate prior on blended corpus:**    Another approach is to incorporate target-domain data when estimating token frequencies, thereby assigning higher prior probabilities to domain-specific terms and preventing their exclusion during filtering. In practice, we mixed Pile-GitHub data with Dolma in equal proportion for prior computation, which needed only an additional 10 minutes of processing. As a result, overall performance improved beyond the original method, while symbolic task performance also increased ("Dolma + Github" among Source column in Table 6).

Table 6: Performance among baselines pre-trained on Dolma.

| | Criteria | Source | Average normalized accuracy | World knowledge | Commonsense reasoning | Language understanding | Symbolic problem solving | Reading compre-hension |
|---|---|---|---|---|---|---|---|---|
| Large (1.5B) model prior-based | mean | Dolma | 8.50 | 9.12 | 10.25 | 7.45 | 11.38 | 4.28 |
| " | stds | " | 8.70 | 7.28 | 10.57 | 9.34 | 12.40 | 3.89 |
| " | mean + stds | " | 9.20 | 9.53 | 11.27 | 10.31 | 11.13 | 3.79 |
| " | " | Dolma + Github | 9.48 | 11.47 | 10.83 | 8.97 | 12.22 | 3.78 |

## D.2    ASSESSING DATA LOSS OF PRIOR-BASED FILTER IN CONTROLLED SETTING

There can be a worry that a highly technical paper with rare terminology or a poem with unusual syntax could be incorrectly filtered. We aim to validate this scenario in a controlled setting. We first consider a scenario where a document with a general structure contains extremely rare terminology. To simulate this, we sampled 1,000 data points (each 512-token length) from the Dolma dataset within the central $\pm 15\%$ range of $\mu$ and gradually injected rare terms into them. Rare terminology was generated by concatenating two tokens ranked in the bottom 10% of the prior distribution (e.g., "prosecromeda", combining two tokens "prosec" and "romeda"). We inserted $n$ of these terms into each text and measured the percentage of these texts

Table 7: $n$ is the number of injected terminology, and inliers is the rate of texts remaining in 25% 75% boundary.

| $n$ | $n\times$ token length | inliers (25% - 75%) |
|---|---|---|
| 1 | 2 | 1.0 |
| 6 | 12 | 1.0 |
| 7 | 14 | 0.98 |
| 8 | 16 | 0.91 |
| 9 | 18 | 0.67 |

classified as outliers ( $\pm 25\%$ threshold). The result is presented in Table 7. Up to seven insertions of rare terms—amounting to 14 tokens (2.7%) within a 512-token block—were required before around 10% of texts were filtered out.

To assess whether seven occurrences are a reasonable upper bound, we examined the typical frequency of topic-specific terminology in real-world text. Specifically, we sampled 10,000 Wikipedia articles, segmented them into 512-token blocks, normalized all text to lowercase, and counted the occurrences of the article title within each block. On average, the title appeared 1.09 times per block, corresponding to an average token length of 3.44. This suggests that even documents intended to explain a given concept are far from being dominated by that terminology. Taken together, these findings indicate that the prior-based filter exhibits strong robustness when handling scenarios involving rare terminology.

# E ADDITIONAL ANALYSIS

## E.1 SENSITIVITY

**Sensitivity to block size** We conduct the following analysis to assess whether outliers remain consistently detected across different block sizes. We randomly concatenated two 512-token blocks to make 1024-token blocks, then trimmed $e/2\%$ from each side to identify outlier samples: $x^{1024}i = x^{512}2i \oplus x^{512}_{2i+1}$, where $i$ denotes the data index in the corpus X, and $x^n$ represents a text block of size n, with $x^{512} \in X^{512}$.

Table 8: Overlaps of outliers between dataset with different block size.

|        | $n = 1024$ | $n = 2048$ |
|--------|-----------|-----------|
| $e = 5$  | 0.7935    | 0.6954    |
| $e = 10$ | 0.8145    | 0.7263    |
| $e = 20$ | 0.8102    | 0.7265    |

If a sample $x_i^{512}$ is classified as a $e\%$ outlier, we then check whether the concatenated block $x_{i//2}^{1024}$ is also classified as an outlier. We repeat the same comparison for $x^{2048}$. As shown in Table 8, outliers of smaller blocks were largely retained as outliers in larger blocks, indicating strong alignment. However, the overlap diminishes as n grows. This is because if a page contains both noisy and (more or equal proportion of) clean content, discarding the entire page may not always be ideal.

**Sensitivity to threshold** Regarding the threshold ratio of outliers, we follow the optimal threshold of the PPL-based filter. Ankner et al. (2024) extensively evaluated PPL-based filtering across different selection rates (25%, 50%, 75%) and concluded that 50% was the most effective. Nevertheless, we explore an additional approach for verification ("elbow" in Table 9). As illustrated in Figure 2, we normalize $\mu$, $\sigma$, rank to a $0 - 1$ scale, and define boundaries where the gradient crosses $-1$ (on both sides), resulting in an 82% threshold of the data. However, the overall performance was lower than the 50% threshold. We also present an experiment with a threshold of 25%, which also showed clearly lower performance. These observations were consistent with the observation of Ankner et al. (2024).

Table 9: Performance of large (1.5B) models on Dolma, under different threshold settings.

| Threshold | Average normalized accuracy | World knowledge | Commonsense reasoning | Language understanding | Symbolic problem solving | Reading comprehension |
|-----------|------------------------------|-----------------|-----------------------|------------------------|---------------------------|------------------------|
| 25%        | 6.21 | 7.03  | 7.06  | 5.99  | 7.36  | 3.57 |
| 50%        | 9.20 | 9.53  | 11.27 | 10.31 | 11.13 | 3.79 |
| 81% (elbow)| 8.79 | 10.04 | 7.67  | 8.97  | 13.04 | 4.21 |

## E.2 CORRELATION BETWEEN PRIOR AND PPL

We present an additional analysis illustrating the correlation between the prior and PPL. Our hypothesis is that this correlation would be more pronounced among outliers, as inlier data exhibit very low variance, making rankings fluctuate easily.

To examine this, we selected the top, middle, and bottom 1k samples based on $\mu$ and computed their average PPL ($E(PPL)$, see Table 10). While the results initially appear to show a highly linear correlation, this is largely driven by extreme outliers within each subset. To mitigate this effect, we recalculated the average after

Table 10: 1000 samples from the top, middle, and bottom ranks.

|          | $E(PPL)$ | $E(PPL*)$ trimmed | $abs(E(PPL*) - M(PPL))$ |
|----------|----------|-------------------|--------------------------|
| $e = 5$  | 1.5      | 1.4               | 33.1                     |
| $e = 10$ | 460.2    | 34.7              | 0.1                      |
| $e = 20$ | 4701.9   | 9.0               | 25.6                     |

trimming the top and bottom five samples from each subset ($E(PPL*)$), and then measured the 1-norm difference from the median PPL: $|E(PPL*) - M(PPL)|$.

The results reveal that subsets centered on $\mu$ correspond to median PPL values, whereas subsets with extreme $\mu$ values (top and bottom) exhibit PPL averages that are substantially distant from the median. This indicates that prior-based rankings effectively identify segments of the data that deviate from the central distribution under PPL.

# F DETAILS ON EXPERIMENTS

## F.1 SCORES FOR EACH BENCHMARK

Table 11: Benchmark performance of large (1.5B) models.

| Model | World knowledge | | | Commonsense reasoning | | | Language understanding | | | |
|---|---|---|---|---|---|---|---|---|---|---|
| | ARC easy | BIG-bench wikidata | TriviaQA | COPA | OBQA | PIQA | HellaSwag | LAMBADA | Winograd | Winogrande |
| no-filter | 8.25 | 2.81 | 0.40 | 0.31 | -4.00 | 15.34 | 1.30 | 6.68 | 12.82 | 3.71 |
| DSIR | 9.65 | 4.42 | 0.47 | 1.47 | 0.53 | 16.00 | 2.70 | 13.43 | 13.55 | -0.71 |
| PPL-based | 11.79 | 8.19 | 0.87 | 2.34 | 0.27 | 19.48 | 4.11 | 16.85 | 9.89 | -1.18 |
| Prior-based (ours) | 12.29 | 6.78 | 1.27 | 1.38 | -0.53 | 20.35 | 5.84 | 18.46 | 14.29 | 2.45 |

| Model | Symbolic problem solving | | | | | | Reading comprehension | | | |
|---|---|---|---|---|---|---|---|---|---|---|
| | BIG-bench algorithms | BIG-bench dyck languages | BIG-bench elementary math QA | BIG-bench operators | GSM8K | SVAMP | LSAT-LR | LSAT-RC | SAT-English | CoQA |
| no-filter | 37.12 | 13.00 | 2.21 | 7.14 | 0.00 | 6.67 | 3.79 | 3.48 | 6.80 | 0.31 |
| DSIR | 39.92 | 13.70 | 2.70 | 5.71 | 0.15 | 1.33 | 3.79 | 4.48 | 6.15 | 1.47 |
| PPL-based | 25.23 | 0.60 | 3.27 | 7.14 | 0.68 | 3.33 | 3.53 | 4.48 | 5.50 | 2.34 |
| Prior-based (ours) | 33.03 | 11.50 | 3.75 | 5.71 | 0.23 | 1.67 | 3.01 | 3.98 | 6.80 | 1.38 |

Table 11 reports the performance of large (1.5B) models on Dolma across different filtering methods. As discussed above, the *prior-based* generally outperforms other baselines or performs comparably to the best baselines.

## F.2 JUSTIFICATION OF EXPERIMENT SCALE

### F.2.1 REGARDING TRAINING TOKEN DURATION

To ensure experimental rigor despite our resource constraints, we followed the experimental setup of a representative paper that we conceptually inherit Ankner et al. (2024). According to this study, the performance gap between baselines is established in the early phases of training. Ankner et al. (2024) reports downstream task performance up to 25 or 50B token duration, and it shows that the performance gaps between baselines observed at 5B token duration remain largely consistent throughout the last stage (Figure 1 of Ankner et al. (2024)). Our work leverages this observation, and this also aligns with our experiment in Figure 7.

Particularly in Ankner et al. (2024), all models are trained with 2× data repetition, and the 5B-token duration in Figure 1 of Ankner et al. (2024) corresponds to 2.5B tokens of data repeated twice. Our experimental setup also uses 3B tokens repeated twice (total 6B), which makes it directly comparable to the 5B-token point in Figure 1 of Ankner et al. (2024).

These trends are consistently observed in other works. li2024datacomp also demonstrates that the ranking of model performance is largely determined early in training and remains consistent. In addition, Figure 1 of diao2025climb clearly shows that the initial performance gaps are established in the early phases.

### F.2.2 REGARDING THE MODEL SIZE

Many representative works on data filtering employ relatively small models around 1.5B parameters, assuming that such a scale is sufficient for reliable evaluation. For example, Ankner et al. (2024) uses 1B and 2B models, while Diao et al. (2025) employs 350M and 1B models. In addition, recent

representative works in this topic (Li et al., 2024b; Penedo et al., 2024; Longpre et al., 2024; Xie et al., 2023b) all conduct experiments with models smaller than 2B. This contrasts with studies focusing on model architectures (e.g., LLaMA, DeepSeek) that compete in complex reasoning performance, which typically require pretraining of large-scale models (e.g., 7B and 70B). A detailed rationale of why small model settings can still maintain experimental rigor in data filtering, is provided below.

### F.3 INDIRECT EVALUATION ON THE COMPLEX REASONING TASK

Certain abilities, like complex reasoning skills, may exhibit emergence in the later stages of training or in larger models. This may be why the reviewers expect experiments at a larger scale. We will provide an explanation and an indirect evaluation regarding this.

#### F.3.1 JUSTIFICATION

Our method is designed to filter noisy text data, which consists of broken sentence structures. If we can assume noise as "data that contains no interpretable information", then learning such data does not contribute to the emergence of complex reasoning. This is because the emergence of complex intelligence arises from the integration of information within the model (Cleveland, 1982), but noise contains no information to contribute to this process. See noise cases filtered by prior-filter in Appendix F; they will remain useless no matter how much the model learns it or how large model learns it.

If we can assume that learning more information (i.e., from corpora with less noise) leads to better complex reasoning skills, the amount of learned information can be evaluated at a smaller scale and with more straightforward benchmarks, which are sufficiently covered in our experiments.

#### F.3.2 INDIRECT EVALUATION

Recently, some research views that reasoning skill is learned directly from the dataset as a part of pattern memorization, rather than emerging as a mere intellectual invention (Wu et al., 2024; Mirzadeh et al., 2024). From this view, if the training dataset contains more data that is similar to the targeted complex task benchmark, the model's performance on that target benchmark would increase accordingly. This is because the model can utilize a larger number of direct templates to solve the task, in a manner of pattern memorization.

From this concept, we mixed the targeted benchmark dataset (e.g., MMLU-pro) into the Dolma corpus and measured how much of this benchmark data is maintained through each filtering method. This simulates a scenario where the training dataset contains data similar to the target benchmark. Therefore, if more benchmark data remain after filtering, we can assume there are higher likelihood that such data (i.e., similar to the targeted benchmark) is preserved in a larger-scale setting, leading to better benchmark performance. Although we can not directly evaluate MMLU-pro performance with a small LM (usually under 7B), this may offer an indirect way to estimate the potential impact of filters on complex task-solving capability.

As a result, we find that the prior-based filter achieves significantly better recognition performance on MMLU-Pro data compared to PPL and random selection. Based on this observation, it is difficult to argue that a model trained with PPL-based filter would achieve a better score than a prior-based filter on MMLU-Pro.

Table 12: The remaining portion of the MMLU-Pro dataset for each filtering method and threshold.

| Threshold (<) | ±0.2 | ±0.25 | ±0.3 | ±0.4 | ±0.45 |
|---|---|---|---|---|---|
| PPL-based | 0.222 | 0.292 | 0.436 | 0.740 | 0.987 |
| Prior-based | 0.510 | 0.629 | 0.743 | 0.931 | 0.990 |
| Random | 0.400 | 0.500 | 0.600 | 0.800 | 0.900 |

We analyze that this might be due to our method's better preservation of "rare but valuable data," as discussed in Appendix B.1. MMLU-Pro is known to contain a lot of logical statements, which likely constitute only a small portion of web corpora (e.g., Dolma). As prior-filter is more sensitive to the rare group (§D), it might have a higher chance of capturing such data.

# G    DETAILS ON BENCHMARKS

Jha et al. (2023) also use the MosaicML evaluation gauntlet to perform evaluations in their work. As such, with explicit permission from the authors, we reproduce their text describing the tasks and task categories in the evaluation gauntlet. The following is from Section D of their paper:

The **World Knowledge** category includes the following datasets:

- **ARC easy**: 2,376 easy four-choice multiple choice science questions drawn from grade 3-9 science exams. (Clark et al., 2018)
- **BIG-bench wikidata**: 20,321 questions regarding factual information pulled from Wikipedia. (Srivastava et al., 2023)
- **TriviaQA**: 3,000 question-answering dataset; clipped all answers to be at most 10 tokens long to improve speed. (Joshi et al., 2017)

The **Commonsense Reasoning** category loosely assesses a model's ability to do basic reasoning tasks that require commonsense knowledge of objects, their properties and their behavior. It includes the following datasets:

- **COPA**: 100 cause/effect multiple choice questions. (Roemmele et al., 2011)
- **OBQA (OpenBook QA)**: 500 four-choice multiple choice questions that rely on basic physical and scientific intuition about common objects and entities. (Mihaylov et al., 2018)
- **PIQA**: 1,838 commonsense physical intuition 2-choice multiple choice questions. (Bisk et al., 2020)

**Language Understanding** tasks evaluate the model's ability to understand the structure and properties of languages and include the following datasets:

- **HellaSwag**: 10,042 multiple choice scenarios in which the model is prompted with a scenario and choose the most likely conclusion to the scenario from four possible options. (Zellers et al., 2019)
- **LAMBADA**: 6,153 passages take from books - we use the formatting adopted by OpenAI's version. (Paperno et al., 2016)
- **Winograd Schema Challenge**: 273 scenarios in which the model must use semantics to correctly resolve the anaphora in a sentence. (Levesque et al., 2012)
- **Winogrande**: 1,267 scenarios in which two possible beginnings of a sentence are presented along with a single ending. (Sakaguchi et al., 2021)

**Symbolic problem solving** tasks test the model's ability to solve a diverse range of symbolic tasks including arithmetic, logical reasoning, algorithms and algebra. These datasets include:

- **BIG-bench algorithms**: 1,320 multiple choice questions. (Srivastava et al., 2023)
- **BIG-bench dyck languages**: 1000 complete-the-sequence questions. (Srivastava et al., 2023)
- **BIG-bench elementary math QA**: 38,160 four-choice multiple choice arithmetic word problems. (Srivastava et al., 2023)
- **BIG-bench operators**: 210 questions involving mathematical operators. (Srivastava et al., 2023)
- **GSM8K**: 1,319 short, free-response grade school-level arithmetic word problems with simple numerical solutions. (Cobbe et al., 2021)
- **SVAMP**: 300 short, free-response grade school-level arithmetic word problems with simple numerical solutions. (Patel et al., 2021)

The **Reading comprehension** benchmarks test a model's ability to answer questions based on the information in a passage of text. The datasets include:

- **LSAT-LR**: 510 passage-based four choice multiple choice questions. (Zhong et al., 2023)
- **LSAT-RC**: 268 passage-based four choice multiple choice questions. (Zhong et al., 2023)
- **SAT-English**: 206 passage-based four choice multiple choice questions. (Zhong et al., 2023)
- **CoQA**: 7,983 passage-based short free response questions. (Reddy et al., 2019)

## H  USAGE OF AI ASSISTANTS

In preparing this manuscript, we relied on AI-powered writing tools to refine sentence flow, fix grammatical mistakes, and improve readability. These assistants were used strictly for language polishing and played no role in shaping the technical content, research design, or experimental work. All scientific concepts, findings, and conclusions presented in this paper were fully developed and written by the researchers. The involvement of AI was limited to editorial support and did not influence the originality or intellectual contributions of the study.

## I  REPRODUCIBILITY STATEMENT

Within the abstract, we included a URL to an anonymous GitHub repository where our code is made available. This repository not only contains the full implementation but also offers detailed guidelines for installation, along with step-by-step instructions on how to perform both training and testing.

## J  FILTERED CASES

In this section, we provide more cases that are classified as extreme outliers by the prior-based criteria.

### J.1  PRIOR STDS($\sigma_d$) BASED

```
'.e.n.i.e..w.s.z.e.l.k.i.c.h..m.a.g.i.c.z.n.y.c.h..o.s.ł
.o.n..j.ą.\n.c.h.r.o.n.i.ą.c.y.c.h.,..w.ł.ą.c.z.a.j.ą.c.
.w..t.o..n.a.s.t.ę.p.u.j.ą.c.e..z.a.k.l.ę.c.i.a.:.\n.ż.e
.l.a.z.n.a..s.k.ÃƒÂª.r.a.,..O.d.p.o.r.n.o.ś.ć..n.a..m.a.
g.i.ę.,..T.a.r.c.z.a..A.r.c.h.o.n.ÃƒÂª.w.,.\n.M.n.i.e.j.
s.z.e..o.d.b.i.c.i.e..c.z.a.r.u.,..K.l.o.s.z..n.i.e.w.r.
a.ż.l.i.w.o.ś.c.i..m.n.i.e.j.s.z.e.j.,.\n.N.i.e.p.o.d.a.
t.n.o.ś.ć..n.a..c.z.a.r.y.,..K.l.o.s.z..n.i.e.w.r.a.ż.l.
i.w.o.ś.c.i.,..M.n.i.e.j.s.z.e.\n.o.d.c.h.y.l.e.n.i.e..c
.z.a'
```
bottom 1

```
'P.A.P.N.-C.A.P.R.N.A.P.R.N.,
BCA.R.N.P.AdministratorB.D.S.B.D.S.,
D.D.S.B.S.C.N.B.Sc.N.C.A.A.C.D.A.C.D.T.C.F.N.C.F.N.P.C.M
.A.C.N.A.C.N.M.C.N.M.T.C.N.P.C.N.S.C.R.N.C.R.N.A.C.R.N.P
.CounselorD.A.D.A./LabTechD.C.D.D.S.D.D.S.-
M.S.D.D.M.D.D.M.D., M.D.D.N.P.D.O.D.O., FACEPD.O.,
M.P.H.D.O., M.S.D.O., ResD.P.M.E.N.P.F.N.P.F.N.P.,-
B.C.F.N.P.-C.G.N.G.N.P,-B.C.J.D.L.C.S.W.L.C.S.W.-
RL.LB.L.P.N.L.P.N.L.V.N.LMHC, LMFTM.A.M.D.M.D.,
Ph.D.M.D., ResM.H.M.M.F.T.M.P.A.-C.M.P.A.-
C.M.P.A.S.M.P.A.S.,
P.A.CM.Rad.M.S.M.S.H.C.M.S.N.M.S.W.MBBS, FACDMBChB,
FRCPCMSN, APRN, NPN.M.T.N.P.N.P.-C.N.T.-C.O.D.O.T.Office
MgrP.A.-C.P.C.P.H.P.'
```
bottom 2

```
'.d. n.d. n.d. n.d. n.d. n.d. n.d. n.d. n.d.\nmax. gain
n.d. n.d. n.d. n.d. n.d. n.d. n.d. n.d. n.d.\nmax. loss
n.d. n.d. n.d. n.d. n.d. n.d. n.d. n.d. n.d.\nTiO2
Median gain/loss n.d. n.d. n.d. n.d. n.d. n.d. n.d.
n.d.\nMAD n.d. n.d. n.d. n.d. n.d. n.d. n.d.\nmax.
gain n.d. n.d. n.d. n.d. n.d. n.d. n.d.\nmax. loss
n.d. n.d. n.d. n.d. n.d. n.d. n.d. n.d.\nBaO 0.004 0.004
n.d. n.d. 0.09 0.02 0.29 0.02 0.62 0.03 n.d. n.d. n.d.
n.d. n.d.\nNiO n.d. n.d. n.d. n.d. n.d. n.d. n.d.
n.d. 0.01 0.01 0.02 0.01 0.02 0.02 n.d. n.d.\nF 0.004
0.004 n.d. n.d. 0.11 0.11 0.75 0.15 n.d. n.d. n.d. n.d.
n.d. n.d. n.d.\nCl 0.004 0.004 n.d. n.d. 0.02 0.01
0.02 0.01 0.03 0.004 n.d. n.d. n.d. n.d. n.d.
n.d.\nAmount of total cations plus OH, n.d'
```
bottom 3

```
'.8.\n8.1. • 8.2.\n9.1. • 9.2. • 9.3. • 9.4. • 9.5. •
9.6. • 9.7. • 9.8. • 9.9. • 9.10. • 9.11. • 9.12. •
9.13.\n10.1. • 10.2. • 10.3. • 10.4. • 10.5. • 10.6. •
10.7. • 10.8. • 10.9. • 10.10. • 10.11. • 10.12. •
10.13. • 10.14. • 10.15. • 10.16. • 10.17. • 10.18. •
10.19. • 10.20.\n11.1. • 11.2. • 11.3. • 11.4. • 11.5. •
11.6. • 11.7. • 11.8. • 11.9.\n12.1. • 12.2. • 12.3. •
12.4. • 12.5. • 12.6. • 12.7. • 12.8. • 12.9. • 12.10. •
12.11. • 12.12. • 12.13. • 12.14. • 12.15. • 12.16. •
12.17. • 12.18. • 12.19.\n13.1. • 13.2. • 13.3. • 13.4.
• 13.5. • 13.6.\n14.1. • 14.2. • 14.3. • 14.4. •
14.5.\n15.1. • 15.2. • 15.3. • 15.4. • 15.5. • 15.6. •
15.7. • 15.8. • 15.9. • 15.10. • 15.11. • 15.12. •
15.13. • 15.14. • 15.15. • 15.16. • 15.17. • 15.18. •
15.19. • 15.20. • 15.21. • 15.22. • 15.23. •
15.24.\n16.1. • 16.2. • 16.3. • 16.4'
```
bottom 4

```
' • p.3. • p.4.\ni.1. • i.2. • i.3. • i.4. • i.5. • i.6.
• i.7. • i.8. • i.9. • i.10. • i.11. • i.12.\nn.1. •
n.2. • n.3. • n.4. • n.5. • n.6. • n.7. • n.8. • n.9. •
n.10.\n1.1. • 1.2. • 1.3. • 1.4. • 1.5. • 1.6. • 1.7. •
1.8. • 1.9. • 1.10. • 1.11. • 1.12. • 1.13. • 1.14. •
1.15. • 1.16. • 1.17. • 1.18. • 1.19. • 1.20. • 1.21. •
1.22. • 1.23. • 1.24. • 1.25. • 1.26. • 1.27. • 1.28. •
1.29.\n2.1. • 2.2. • 2.3. • 2.4. • 2.5.\n3.1. • 3.2. •
3.3. • 3.4. • 3.5. • 3.6. • 3.7. • 3.8. • 3.9. • 3.10. •
3.11. • 3.12. • 3.13. • 3.14. • 3.15. • 3.16. • 3.17. •
3.18. • 3.19. • 3.20. • 3.21.\n4.1. • 4.2. • 4.3.• 4.4.•
4.5.\n5.1. • 5.2. • 5.3. • 5.4. • 5.5. • 5.6. • 5.7. •
5.8. • 5.9. • 5.10. • 5.11.\n6.1. • 6.2. • 6.3. • 6.4. •
6.5. • 6.6. • 6.7. • 6'
```
bottom 5

```
'. Candle. Wax. Wick. Lit. Cloud. Smoke. Smoldering.
Embers. Blaze. Fire. Firebug. Firefighter. Fire
extinguisher. Flammable. Volatile. Combustible.
Incendiary. Implosion. Tamp. Explosive. Boom. Blast.
Bullet. Gunpowder. Nitroglycerin. Dynamite. A fuse.
Pyrotechnics. Fireworks. Firecracker. Skyrocket.
Meteoric. Revolve. Orbit. Satellite. Trajectory.
Fleeting. Barrage. Volley. Uzi. Canon. Consecutive.
Streak. Rapid fire. Incoming. Archery. Quiver. Arrow.
Rocket. Missile. Silo. Torpedo. Launch. Fling. Catapult.
Slingshot. Propulsion. Thrust. Throw. Pitch. Pitcher.
Fastball. Hardball. Curveball. Pinball. Bounce. Bump.
Paramotor. Paragliding. Airspace. Supersonic. Jet.
Helicopter. Tilt. Torque. Propeller. Spin. Turn. Rotate.
Whirl. Swirl. Twist. Twister. Tornado. Whirlwind.
Unleash. Target. Bull's-eye. Marksman. Aim. Point.
Hotshot. Hop. Bunny. Rabbit. Hare. Birds. Metabolism.
Mitochondria. Carbohydrates. Calories. Tomato. Royal
jelly. Honey. Hornets. Wasps. Buzzing. Humming. Hive.
Swarm. Sting. Scratch. Itch. Rash. Skin. Flesh.
Complexion. Dermatologist. Acne. Inflammation. Tingle.
Injection. Red. Redhead. Nozzle. Siphon. Bladder.
```
bottom 10

```
'.devtools.build.lib.shell.JavaSubprocessFactory;\nimport
com.google.devtools.build.lib.shell.SubprocessBuilder;\nimport
com.google.devtools.build.lib.shell.SubprocessFactory;\nimport
com.google.devtools.build.lib.util.AbruptExitException;\nimport
com.google.devtools.build.lib.util.CustomExitCodePublisher;\nimport
com.google.devtools.build.lib.util.CustomFailureDetailPublisher;\nimport
com.google.devtools.build.lib.util.DebugLoggerConfigurator;\nimport
com.google.devtools.build.lib.util.DetailedExitCode;\nimport
com.google.devtools.build.lib.util.ExitCode;\nimport
com.google.devtools.build.lib.util.InterruptedFailureDetails;\nimport
com.google.devtools.build.lib.util.LoggingUtil;\nimport
com.google.devtools.build.lib.util.OS;\nimport
com.google.devtools.build.lib.util.Pair;\nimport
com.google.devtools.build.lib.util.ProcessUtils;\nimport
com.google.devtools.build.lib.util.TestType;\nimport
com.google.devtools.build.lib.util.ThreadUtils;\nimport
com.google.devtools.build.lib.util.io.OutErr;\nimport
com.google.devtools.build.lib.vfs.FileSystem;\nimport
com.google.devtools.build.lib.vfs.FileSystemUtils;\nimport
```
bottom 100

```
' Amend, A. Mozeika, E. Steltz, M. R. Zakin, H. Lipson, H. M.\nJaeger, Proc. Natl.
Acad. Sci. U. S. A. 2010, 107, 18809.\nM. Wang, G. M. Whitesides, Proc. Natl. Acad.
Sci. U. S. A. 2011, 108, 20400.\nScience 2012, 337, 828.\n[9] R. V. Martinez, A. C.
Glavan, C. Keplinger, A. I. Oyetibo, G. M. Whitesides, Adv. Funct.\nMater. 2014, 24,
3003.\n[10] R. V. Martinez, C. R. Fish, X. Chen, G. M. Whitesides, Adv. Funct. Mater.
2012, 22, 1376.\n[11] R. F. Shepherd, A. A. Stokes, J. Freake, J. Barber, P. W.
Snyder, A. D. Mazzeo, L.\nCademartiri, S. A. Morin, G. M. Whitesides, Angew. Chem.
2013, 125, 2964.\nR. Nunes, G. M. Whitesides, R. J. Wood, in 2014 IEEERSJ Int. Conf.
Intell. Robots Syst.\nIROS 2014, 2014, pp. 561566.\nG. M. Whitesides, R. J. Wood,
Science 2015, 349, 161.\n[14] B. Mosadegh, P. Polygerinos, C. Keplinger, S.
Wennstedt, R. F. Shepherd, U. Gupta, J.\nShim, K. Bertoldi, C. J. Walsh, G. M.
Whitesides, Adv. Funct. Mater. 2014, 24, 2163.\n[15] M. T. Tolley, R. F. Shepherd, B.
Mosadegh, K. C. Galloway, M. Wehner, M. Karpelson, R.\nJ. Wood, G. M. Whitesides,
Soft Robot. 2014, 1, 213.\n[16] L. Margheri, C. Laschi, B. Mazzolai, Bioinspir.
Biomim.'
```
bottom 500

```
'1000 717.000000 0.000000 136.000000 60.000000\n7.8500 44.8400 55.1000 727.000000
0.000000 136.000000 60.000000\n7.8600 44.8400 55.1500 733.000000 0.000000 136.000000
60.000000\n7.8700 44.8400 55.1500 721.000000 0.000000 136.000000 60.000000\n7.8800
44.8400 55.1500 724.000000 0.000000 136.000000 60.000000\n7.8900 44.8400 55.1500
722.000000 0.000000 136.000000 60.000000\n7.9000 44.9200 55.2200 722.000000 0.000000
136.000000 60.000000\n7.9100 45.0000 55.3000 720.000000 0.000000 136.000000
60.000000\n7.9200 45.1400 55.3800 722.000000 0.000000 136.000000 60.000000\n7.9300
45.3700 55.4900 721.000000 0.000000 136.000000 60.000000\n7.9400 45.6500 55.6200
724.000000 0.000000 136.000000 60.000000\n7.9500 45.9900 55.8200 724.000000 0.000000
138.000000 60.000000\n7.9600 46.5000 56.0300 724.000000 0.000000 136.000000
61.000000\n7.9700 47.1000 56.2600 727.000000 0.000000 136.000000 61.000000\n7.9800
47.9500 56.5700 737.000000 0.000000 138.000000 61.000000\n7.9900 49.0300 56.9400
760.000000 0.000000 140.000000 62.000000\n8.0000 50.2700 57.3400 771.000000 0.000000
140.000000 62.000000\n8.0100 51.8700 57.8000 788.000000 0.000000 141.000000
63.000000\n8.0200 53.6500 58.3900 790.000000 0.000000 143.000000 63.000000\n8.0300
55.5400 59.0700 799.000000 0.000000 144.000000 62.000000\n8.0400 57.4300 59.6300
809.000000 0.000000 146'
```
bottom 1000

```
'.49\t1268.33\n462.69\t1255.34\n462.90\t1264.70\n463.11\t1255.02\n463.31\t1269.92\n46
3.52\t1258.15\n463.73\t1256.04\n463.94\t1259.95\n464.14\t1257.80\n464.35\t1253.62\n46
4.56\t1260.20\n464.76\t1265.81\n464.97\t1262.90\n465.18\t1255.32\n465.38\t1256.90\n46
5.59\t1267.92\n465.80\t1264.13\n466.00\t1263.35\n466.21\t1258.56\n466.42\t1259.44\n46
6.62\t1266.71\n466.83\t1258.23\n467.04\t1265.19\n467.24\t1255.65\n467.45\t1265.99\n46
7.66\t1266.02\n467.86\t1254.33\n468.07\t1262.00\n468.28\t1256.32\n468.48\t1257.82\n46
8.69\t1267.04\n468.90\t1254.89\n469.10\t1263.70\n469.31\t1257.70\n469.52\t1265.11\n46
9.72\t1249.30\n469.93\t1263.64\n470.14\t1252.37\n470.34\t1263.56\n470.55\t1253.73\n47
0.76\t1256.43\n470.96\t1252.31\n471.17\t1263.13\n471.38\t1251.80\n471.58\t1256.06\n47
1.79\t1254.81\n472.00\t1264.05\n472.20\t1249.85\n472.41\t1261.47\n472.62\t1257.08\n47
2.82\t1262.10\n473.03\t1257.41\n473.24\t1264.17\n473.44\t1255.06\n473.65\t1261.73\n47
3.85\t1252.93\n474.06\t1263.31\n'
```
bottom 10000

Figure 8: Bottom-$n$ ranked data based on prior stds ($\sigma_d$).

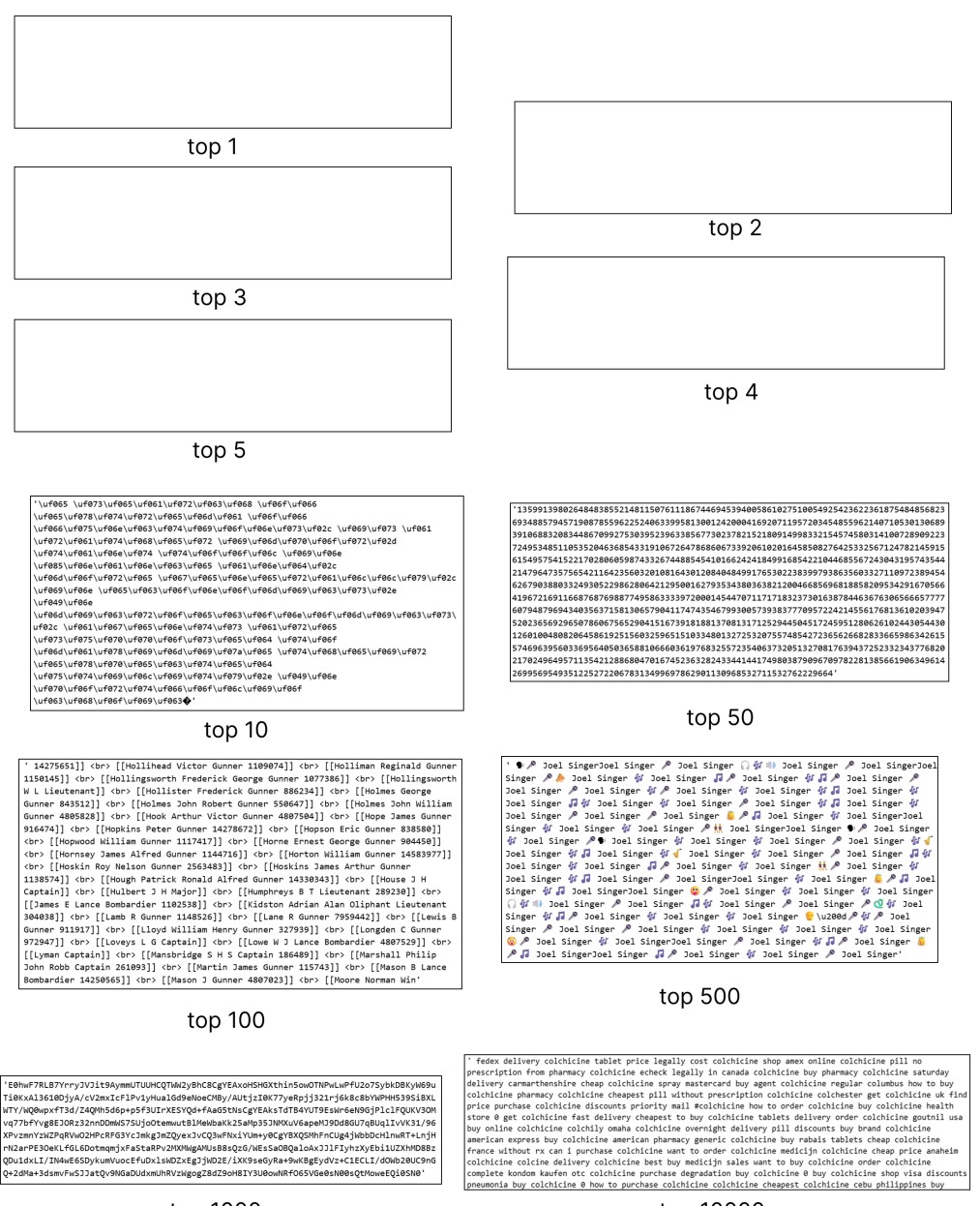

Figure 9: Top-$n$ ranked data based on prior stds ($\sigma_d$). For the top 1–5 examples, the whitespace contains special characters other than "/s".

## J.2 PRIOR MEAN($\mu_\mathrm{d}$)-BASED

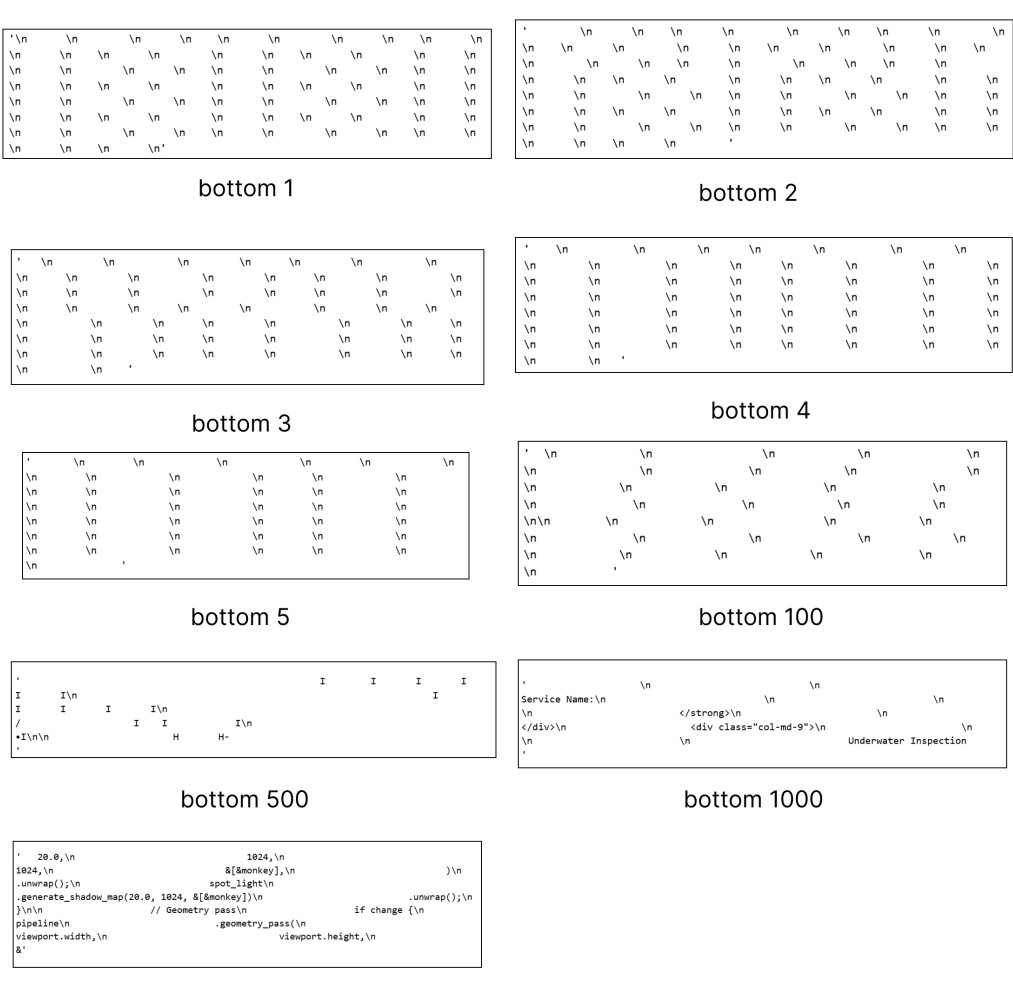

Figure 10: Bottom-$n$ ranked data based on prior mean ($\mu_\mathrm{d}$).

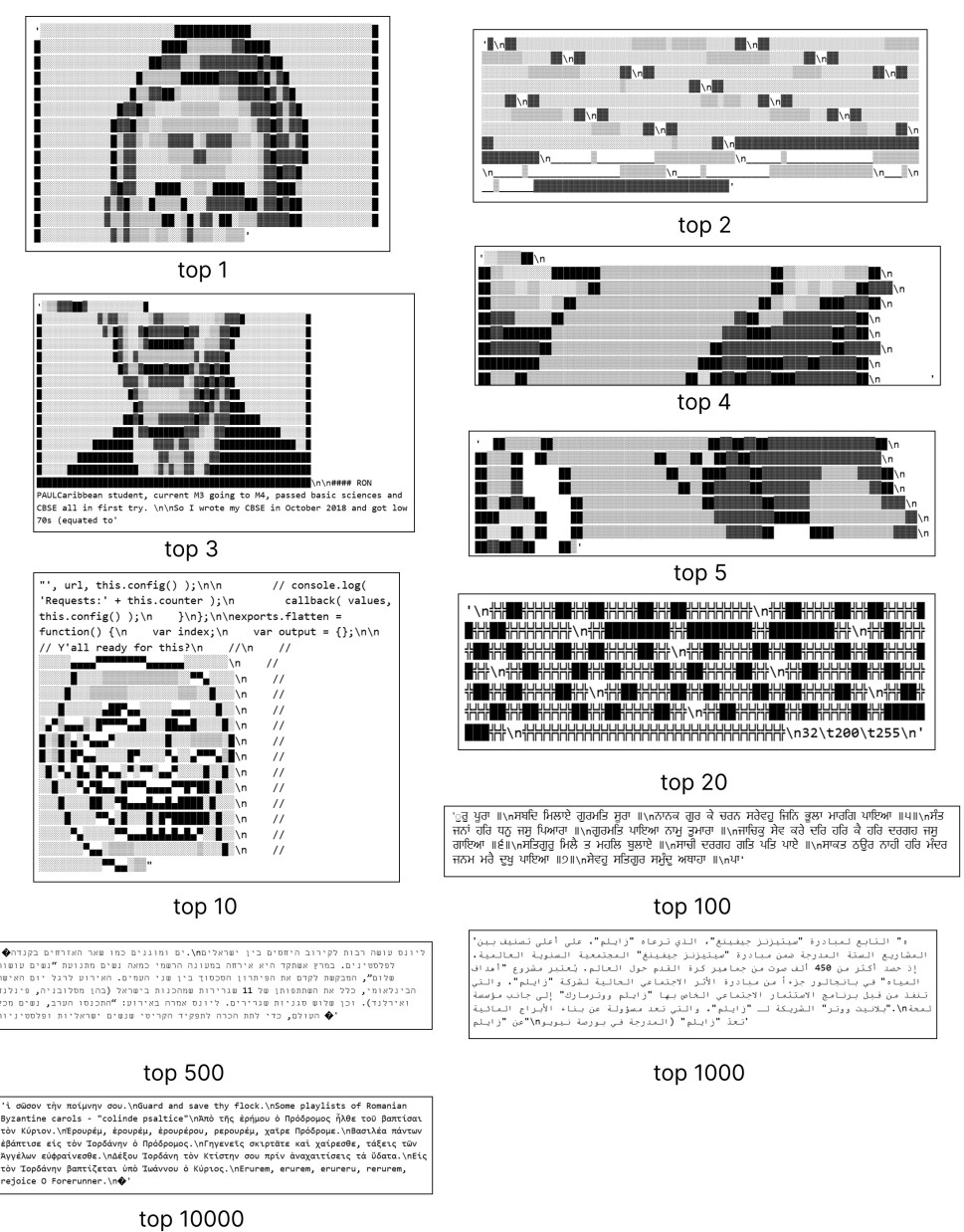

Figure 11: Top-$n$ ranked data based on prior mean ($\mu_\mathrm{d}$).

