# OpenReview forum: "Prior-based Noisy Text Data Filtering: Fast and Strong Alternative For Perplexity"
_ICLR.cc/2026/Conference — ICLR 2026 Poster_

### Official Review · Reviewer_8chv · 2025-10-29

**Soundness:** 3
**Presentation:** 3
**Contribution:** 3
**Rating:** 6
**Confidence:** 4

**Summary:**

The paper proposes a prior-based data filtering approach that replaces perplexity-based filtering with a simple frequency-based proxy. By leveraging corpus-level token priors (term frequencies) and their document-level mean and variance, the method identifies noisy or ill-formed data efficiently. It achieves comparable or better downstream performance than PPL-based methods while being 1000x faster.

**Strengths:**

- Efficient and elegant alternative to PPL filtering.
- Strong empirical results across 20 benchmarks and two model sizes.
- Grounded in interpretable linguistic insights (lexical density, word frequency).
- Scalable and robust, it extends to code and math data.

**Weaknesses:**

- Results mostly on English and related datasets; generalization to other scripts unclear: The experiments are focused almost entirely on English or similar languages, so it’s hard to tell how well the method would hold up for other scripts or morphologically rich languages. Since the approach depends on token frequencies, it might behave very differently for languages with complex segmentation or compounding (like Japanese or Turkish). Even a small multilingual check would help back up the broader claims.
- Some claims (e.g., automatic "learnability" detection) lack quantitative rigor: The paper argues that the method can automatically detect “learnable” data, but this idea isn’t really supported by clear evidence. There aren’t concrete metrics or controlled experiments showing that the filtered data is actually easier for a model to learn from. As it stands, the claim feels more intuitive than proven.
- May remove low-frequency, high-value data: Because the method penalizes documents with rare token patterns, it could unintentionally throw away useful or unique data like text from small domains, rare languages, or creative writing. This might reduce the diversity of the pretraining corpus, especially in the long tail where valuable but infrequent patterns live. A short analysis of this trade-off would make the results more convincing.
- Some sections read more as post-hoc rationalization (linguistic grounding) than rigorous analysis.
- Limited novelty from a machine learning standpoint

**Questions:**

-How sensitive are results to tokenizer choice or token granularity?
- Could the method over-filter rare but meaningful content?
- How does this interact with deduplication or semantic filtering pipelines?

---

> ### Author Response · Authors · 2025-11-23
> **Official Comment to Reviewer 8chv <1>**
>
> Thank you so much for raising very interesting points! We will make sure to include the discussions in the revision or the final version. Thank you very much for your time!
>
> ---
>
> $\color{white}$
>
> **[W1] Results mostly on English and related datasets. It might behave very differently for languages with complex segmentation or compounding (like Japanese or Turkish).**
>
> = **(1) We already proved generalization in an extremely distinct type of language (e.g., symbolic language)**
>
> Our method is supposed to work with any language system composed of function words and content words, which includes most natural languages (e.g., Chinese, Japanese, Turkish). Symbolic languages (e.g., codes) are a more worrisome case, as they are more different from English, and it has not been verified whether linguistic observation of natural language applies to them. **However, we already conducted experiments on this extreme case (**§**4.2)**, and proved our method still generalizes. We can expect that it will generalize to other natural languages more similar to English.
>
> $\color{white}$
>
> **(2) Our method generalizes to other natural languages**
>
> If you are curious whether our method works for natural languages other than English, the Chinese–English experiment (§3.4.2) provides evidence for this.
>
> Suppose that the prior-based filter works only for English but not for Chinese (as you suspect), then our filter will classify all Chinese data as noise. Then, in Figure 5, at the point where Chinese and English are mixed at a 10:100 ratio (at any ratio), nearly 100% of the Chinese data should be classified as outliers. However, the actual results show that most of the Chinese data were classified as inliers, only if this language group takes more than 10% of the corpus. Therefore, the prior-based filter works for Chinese as well.
>
> We also conducted the same experiment on Turkish, as you mentioned. And the results were consistent with those of the Chinese–English experiment. We provide an explanation below.
>
> $\color{white}$
>
> Etable 1.  The same setting as in Figure 5 of the paper. Proportion of Turkish data classified as outliers, after mixing Turkish and English data at a ratio of a : 100. Outliers are the
> top and bottom 5% of µd.
>
> | Size of Turkish data compared to En (a) | Proportion of outliers (10%) |
> | --- | --- |
> | 1 | 0.5036 |
> | 5 | 0.2778 |
> | 10 | 0.2448 |
> | 20 | 0.1984 |
> | 100 | 0.1337 |
>
> $\color{white}$
>
> **#. Explanation for our method’s generalization**
>
> Our method can generalize, because tokenizers split words into morphemes based on their frequency, no matter it is how compounded. Based on the following two properties, our approach generalizes to all natural languages.
>
> **(1) No matter how compound a language is, it can be separated into function morphemes and content morphemes** [2,3].
>
> For instance, the Turkish compounding word *evlerimizden* (‘from our homes’) decomposes into *ev* (content morpheme) + *ler* + *imiz* + *den* (grammatical morphemes).
>
> These function and content words follow the frequency rule (i.e., function words tend to have higher frequency). For example, in Turkish, the function word “ler” appears across many different nouns (e.g., *evler*, *şehirler,* *öğrenciler* ), resulting in higher term frequency.
>
> The same phenomenon occurs even when dealing with English. For example, a compound word such as *childish* is typically segmented into *child* and *-ish*. The suffix *-ish* appears across many different content words,  and thus behaves similarly to a high-frequency function morpheme.
>
> $\color{white}$
>
> **(2) The tokenizer also splits words into morphemes based on frequency.**
>
> Most tokenizers (e.g., BPE, WordPiece, SentencePiece) have tendency to merge characters with high co-occurrence probability into a single token (e.g., “i” + “n” → “in”). This is because most tokenizers, either directly or approximately, follow the Minimum Description Length (MDL) [4] objective, which aims to minimize both the vocabulary size and the length of representation. For example, the token *“ler”* frequently co-occurs with various words, so assigning it as an independent token is more efficient (i.e., covering a wider range of expression with limited vocab) than assigning *“şehirler”* and *“evler”* as separate tokens.
>
> Combining together, most of natural languages can be segmented into function and content components, and most tokenizers effectively reflect this distinction, even in morphologically compounding languages.

---

> ### Author Response · Authors · 2025-11-23
> **Official Comment to Reviewer 8chv <2>**
>
> **[W2] Automatic "learnability" detection**
>
> =  It seems that the definition of *learnability* was misdelivered. What you referred to seems to correspond to “difficulty,” whereas our intended meaning is closer to “extractability”. What we mean by *learnability* is **independent of the intrinsic difficulty of the data**. Even identical data can have different levels of learnability depending on how frequently similar instances appear in the corpus.
>
> For example, in English, sentences like *“Tom is a boy”* and *“James is a student”* allow the model to learn the grammatical role of the word “is” and the sentence template around it.  Thus, the more such similar patterns appear in the corpus, the higher the learnability becomes. In contrast, if there is only a single instance, such as *“Tom is a boy,”* the model cannot infer the grammatical role of *“is”.*  In this case, the information can not teach anything to the model (thus unlearnable), **so it is nothing more than a noise.**
>
> Therefore, the notion of *learnability* we introduce is directly tied to the **portion** of a similar family of data in the corpus. Our experimental setup reflects this relationship explicitly: when the portion of a given data type is small, it is classified as an outlier, and as the portion increases, it becomes classified as an inlier.

---

> ### Author Response · Authors · 2025-11-23
> **Official Comment to Reviewer 8chv <3>**
>
> **[W3, Q2]  Preservation of low-frequency, high-value data**
>
> **= #. Our method is robust in preserving rare and valuable data.**
>
>  Loss of rare but valuable data is an inherent cost paid by all filtering methods, and reducing this cost remains one of the goals in this field. About this issue, we already **very thoroughly discussed in §B.1 (referred from** §**4, line 452).**
>
> $\color{white}$
>
> **For your convenience, we will briefly repeat here.**
>
>  The aspect you mentioned has been partially measured through benchmarks, where our baselines demonstrate relative strengths. For example, the scenario you described can be observed in the performance on symbolic task benchmarks within Dolma. Dolma primarily consists of web data ( Common Crawl 74.6%, Table 1), but contains a small amount of code and math scripts (Stack 13.4%), which fits the description of “rare and valuable data”.
>
> In Table 2, PPL-based models underperform compared to the no-filter baseline on these symbolic tasks, revealing the trade-off problem of the filtering method. However, our method achieves higher performance than the PPL-based approach, indicating its superior ability to leverage such high-value minority data.
>
> $\color{white}$
>
> **Nevertheless, to address your concern, we introduce two additional methods for a better trade-off.**
>
> (1) Using only stds:
>
> While the mean reflects the average frequency of tokens, the stds calculate the variation among token frequencies. Therefore, it is more suitable for identifying well-formed text written in low-frequency languages. Empirically, the stds-only filter outperforms both the baselines overall and the stds+mean filter on symbolic tasks. (ETable1)
>
> (2) Calculate prior on blended corpus:
>
> Our second strategy is to incorporate target-domain data when estimating token frequencies, thereby assigning higher prior probabilities to domain-specific terms and preventing their exclusion during filtering. In practice, we mixed Pile-GitHub data with Dolma in equal proportion for prior computation, which needed only an additional 10 minutes of processing.
>
> As a result, overall performance improved beyond the original method, while symbolic task performance also increased. (ETable1)
>
> $\color{white}$
>
> ETable 2:  Performance of large (1.5B) models on Dolma. The *Prior (Dolma + GitHub)* setting uses both std and mean.
>
> |  | Avg | World knowledge | Commonsense reasoning | Language understanding | Symbolic problem solving | Reading comprehension |
> | --- | --- | --- | --- | --- | --- | --- |
> | no-filter | 5.78 | 5.52 | 0.44 | 6.14 | 13.22 | 3.59 |
> | PPL-based | 8.22 | 9.98 | 11.91 | 7.34 | 7.91 | 3.96 |
> | Prior (stds + mean) | 9.20 | 9.53 | 11.27 | 10.31 | 11.13 | 3.79 |
> | Prior  (stds) | 8.70 | 7.28 | 10.57 | 9.34 | 12.40 | 3.89 |
> | Prior (dolma + github) | 9.48 | 11.57 | 10.83 | 8.97 | 12.27 | 3.78 |
>
> $\color{white}$
>
> **#. What if rare terminology is used? (e.g., technical paper)**
>
> We also already validated this scenario in a controlled setting in §B.1.2.
>
> We first consider a scenario where a document with a general structure contains extremely rare terminology. To simulate this, we sampled 1,000 data points (each  512-token length) from the Dolma dataset within the central ±15% range of $\mu$ and gradually injected rare terms into them. Rare terminology was generated by concatenating two tokens ranked in the bottom 10% of the prior distribution (e.g., “prosecromeda”, combining two tokens “prosec” and “romeda”). We inserted $n$ of these terms into each text and measured the percentage of these texts classified as outliers ( ±25% threshold). The result is as follows (ETable 2)
>
> $\color{white}$
>
> ETable 3: $n$ is the number of injected terminology, and inliers is the rate of texts remaining in 25%~75% boundary.
>
> | n | n * token length | inliers (25%~75%) |
> | --- | --- | --- |
> | 1 | 2 | 1.0 |
> | 6 | 12 | 1.0 |
> | 7 | 14 | 0.98 |
> | 8 | 16 | 0.91 |
> | 9 | 18 | 0.67 |
>
> $\color{white}$
>
> Up to seven insertions of rare terms—amounting to 14 tokens (2.7%) within a 512-token block—were required before around 10% of texts were filtered out.
>
> $\color{white}$
>
>  **To assess whether seven occurrences are a reasonable upper bound**, we examined the typical frequency of topic-specific terminology in real-world text. Specifically, we sampled 10,000 Wikipedia articles, segmented them into 512-token blocks, normalized all text to lowercase, and counted the occurrences of the article title (i.e., subject entity) within each block.
>
> On average, the title appeared 1.09 times per block, corresponding to an average token length of 3.44. This suggests that even documents intended to explain a given concept are far from being dominated by that terminology. Taken together, these findings indicate that the prior-based filter exhibits strong robustness when handling scenarios involving rare terminology.

---

> ### Author Response · Authors · 2025-11-23
> **Official Comment to Reviewer 8chv <4>**
>
> **[W4,5] Novelty and post-hoc rationalization**
>
> = Regarding the novelty: We introduce a perspective in which frequency serves as a representation of **function words** and **content words.** To the best of our knowledge, applying this concept to noise filtering is novel. Based on your earlier comments [W3, Q2], we are concerned that you misunderstood our method as being the same as a traditional frequency-based approach (e.g., n-gram), which just filters out text with low-frequency contents. If that is not the case, we would appreciate it if you could clearly provide a reference that presents a concept similar to ours.
>
> Regarding the rationalization: We additionally provide a theoretical guarantee that, only under this linguistic assumption, (1) the prior-based metric serves as a noise detector and (2) how our prior-based metric exhibits aligned behavior with PPL in noise detection. Therefore we proved it is necessary rationalization, not a post-hoc justification. If this is the point of your concern, please refer to our response to **HKmF [W1, W4]**. We apologize for not including the full details here, as we did not want to occupy excessive space if this was not your intended concern.
>
> ---
>
> $\color{white}$
>
> **[Q1] How sensitive are results to tokenizer choice or token granularity?**
>
>
> $\color{white}$
>
> ETable 4. Performance of large (1.5B) models on Dolma, under different tokenizers.
>
> | Tokenizer | Avg | World knowledge | Commonsense reasoning | Language understanding | Symbolic problem solving | Reading comprehension |
> | --- | --- | --- | --- | --- | --- | --- |
> | GPT2 | 9.20 | 9.53 | 11.27 | 10.31 | 11.13 | 3.79 |
> | LLaMA-3 | 9.39 | 9.54 | 11.16 | 10.78 | 11.86 | 3.64 |
> | T5-small | 8.11 | 8.59 | 7.43 | 7.95 | 12.32 | 4.22 |
>
> $\color{white}$
>
> = We clarify that our draft already included the comparison with two additional tokenizers ( **§B.2, refered from §4.1, line 453**): LLaMA-3-8B (UTF-8-based, vocab size 128K), and T5-small
> (SentencePiece-based, vocab size 32K). GPT-2 tokenizer (mainly used in our paper) has a vocab size of 50K.
>
> We conducted an experiment to determine which tokenizer yields better performance, and the result is reported in Table 4. All tokenizers showed the advantage of filtering, while T5 exhibited slightly lower performance gain. We propose two possible interpretations: (1) There may exist a threshold of optimal granularity, which lies between 32K and 50K. (2) Using the tokenizer paired with the model may offer stability.
>
> Nevertheless, as demonstrated, priors can be computed even with tokenizers mismatched to the model. If an optimal tokenizer exists, it can be freely adopted at any time.
>
> ---
>
> $\color{white}$
>
> **[Q3] How does this interact with deduplication or semantic filtering pipelines?**
> = Dolma [1] corpus itself incorporates multiple filtering stages—including deduplication and semantic filtering pipelines.  Our method achieves substantial performance gains on top of this, proving its compatibility when combined.
>
> < Filters applied in Dolma >
>
> - Heuristic rule : adopted from rules used in C4, Gopher.
> - Language filter (classifier-based) :  Using fastText to select English data.
> - Content filter (classifier-based) : Using fastText trained on the Jigsaw Toxic Comment dataset to filter out toxic documents
> - Deduplication (rule-based) : duplication regarding an exact match of URL, document, or paragraph.
>
> $\color{white}$
>
> ---
>
> Thank you very much for your time!
>
> [1] Luca Soldaini et al., Dolma: An Open Corpus of Three Trillion Tokens for Language Model Pretraining Research, ACL 2025
>
> [2] Stromswold, Karin. "The Nature and Acquisition of Lexical and Functional Categories." *ms., Rutgers University, New Brunswick, New Jersey* (1994).
>
> [3] Croft, William. "Typology and universals." *The handbook of linguistics* (2017): 39-55.
> [4] Hou, Jue, et al. Effects of sub-word segmentation on performance of transformer language models,  EMNLP 2023

---

### Official Review · Reviewer_edXW · 2025-10-31

**Soundness:** 2
**Presentation:** 2
**Contribution:** 2
**Rating:** 2
**Confidence:** 3

**Summary:**

This paper introduces a prior-based data filtering method as a simple, fast, and powerful alternative to perplexity (PPL) for cleaning massive web corpora used in LLM pretraining. Inspired by linguistic insights on lexical density, the method estimates token priors using corpus-level term frequency and filters documents based on the mean and standard deviation of these priors, requiring no model inference. Despite its simplicity, the prior-based filter achieves state-of-the-art average performance, outperforming PPL-based filtering across 20 downstream benchmarks. The proposed approach is also over 1000 times faster than PPL filtering and demonstrates robust applicability to multilingual corpora and symbolic languages like code and math.

**Strengths:**

* This paper proposes a simple yet effective data filtering method that achieves efficient information filtering using only statistical features.
* Experiments show that the method is not only fast but also outperforms PPL-based approaches in terms of performance.
* The authors demonstrate the generality of the method, showing that it can be applied to text, code, and math data.

**Weaknesses:**

* The experiments are conducted only on a 3B-scale dataset, which is too small to be convincing. I believe the authors should use a larger dataset (e.g., over 100B) and test on more complex tasks to better demonstrate the superiority of the proposed data filtering method.
* In terms of writing, Chapter 2 includes excessive discussion of basic NLP concepts such as word frequency. I suggest that the authors reduce such background explanations or move them to the appendix.
* The authors only test on unigrams and do not extend the analysis to n-grams. I believe they should further discuss the filtering effectiveness for n-grams.
* The paper lacks experiments on the choice of filtering thresholds and filtering ratios. I suggest that the authors conduct additional experiments on these two hyperparameters to determine the optimal settings.

**Questions:**

* Have the authors considered combining different methods to achieve better filtering performance?
* Is there any scaling experiment on data filtering? The authors should test with larger data proportions and model sizes to fully demonstrate the effectiveness of the filtering approach.
* Why did the authors choose word frequency instead of n-grams or more complex statistical metrics? Previous work in statistical NLP has shown that n-grams are often more effective than using word frequency alone.
* Is there a detailed experiment on the filtering ratio to determine the optimal filtering threshold?
* Would different tokenizers significantly affect the filtering results? Moreover, can the filtering method remain effective across different tokenizers?

---

> ### Author Response · Authors · 2025-11-23
> **Official Comment to Reviewer edXW <1>**
>
> Thanks so much for participating in the review process. While many of the questions are constructive, some appear to stem from misunderstandings of our method **[W3, Q3, W2]** or from not fully reading parts that are already sufficiently explained in the paper **[W4, Q1, Q4, Q5]**. We kindly ask for reading the rebuttal and our paper carefully. All other meaningful discussions will be incorporated into the final version.
>
> ---
>
> $\color{white}$
>
> **[W1, Q2] Request for experiment with larger scale of dataest, model size., and more complex task.**
>
> = [Summary] First, we clarify that our experimental setting maintains rigor in light of observations from previous studies. We also discuss that this scale is sufficient to demonstrate the noise-filtering capability of our method. Nevertheless, to address your concern, we additionally conducted experiments on a larger scale, and also provided indirect evidence showing our method’s advantage in complex reasoning skills.
>
> **#. Experiments on an extended scale**
>
> ETable 1.  Performance of models trained on Dolma. 12B token duration (1.5B model).
>
> |  | Avg | World knowledge | Commonsense reasoning | Language understanding | Symbolic problem solving | Reading comprehension |
> | --- | --- | --- | --- | --- | --- | --- |
> | PPL-based | 9.54 | 12.01 | 13.21 | 9.39 | 8.54 | 4.57 |
> | Prior-based | 10.51 | 11.43 | 12.92 | 11.21 | 12.73 | 4.28 |
> ---
>
> $\color{white}$
>
> We provide additional experiments of the core baselines (prior- and PPL-based) on an extended scale. We additionally trained a checkpoint of 1.5B on an additional subset of Dolma, to be a total of 12B token duration (2 times repetition of 6B). The results were largely consistent with the previous one (Table 2), demonstrating the stable performance of our method.
>
> $\color{white}$
>
> **#. Justification of our scale**
>
> To ensure experimental rigor despite our resource constraints, we followed the experimental setup of a representative paper that we conceptually inherit [1]. We already provided this justification in Appendix C.2.
>
> According to this study, the performance gap between baselines is established early phases of training. [1] reports downstream task performance up to 25 or 50B token duration, and it shows that the performance gaps between baselines observed at 5B token duration remain largely consistent throughout the last stage (Figure 1 of [1]). Our work leverages this observation, and this also aligns with our new experiment in **ETable1.**
>
> Particularly in [1], all models are trained with 2× data repetition, and the 5B-token duration in Figure 1 of [1] corresponds to 2.5B tokens of data repeated twice. Our experimental setup also uses 3B tokens repeated twice (total 6B), which makes it directly comparable to the 5B-token point in Figure 1 of [1].
>
> These trends are consistently observed in other works. [2] also demonstrates that the ranking of model performance is largely determined early in training and remains consistent. In addition, Figure 1 of [3] clearly shows that the initial performance gaps are established in the early phases.
>
> **Our model size remains rigorous compared to previous works**
>
> Regarding the model size, many representative works on data filtering employ relatively small models around 1.5B parameters, assuming that such a scale is sufficient for reliable evaluation. For example, [1] uses 1B and 2B models, while [3] employs 350M and 1B models. In addition, recent representative works in this topic [9, 10, 11, 12] all conduct experiments with models smaller than 2B. This contrasts with studies focusing on model architectures (e.g., LLaMA, DeepSeek) that compete in complex reasoning performance, which typically require pretraining of large-scale models (e.g., 7B and 70B). A detailed rationale of why small model settings can still maintain experimental rigor in data filtering, is provided below.
>
> $\color{white}$
>
> **#. Indirect estimation of the potential advantage in the complex reasoning task**
>
> Certain abilities, like complex reasoning skills, may exhibit emergence in the later stages of training or in larger models. This may be why the reviewers expect experiments at a larger scale. We will provide an explanation and an indirect evaluation regarding this.
>
>
> **(1) Justification**
>
>    ((((continue to the next page))))

---

> ### Author Response · Authors · 2025-11-23
> **Official Comment to Reviewer edXW <2>**
>
> **(1) Justification**
>
> Our method is designed to filter noisy text data, which consists of broken sentence structures. If we can assume noise as "data that contains no interpretable information", then learning such data does not contribute to the emergence of complex reasoning. This is because the emergence of complex intelligence arises from the integration of information within the model [7], but noise contains no information to contribute to this process. See noise cases filtered by prior-filter in Appendix F; they will remain useless no matter how much the model learns it or how large model learns it.
>
> If we can assume that learning more information (i.e., from corpora with less noise) leads to better complex reasoning skills, the amount of learned information can be evaluated at a smaller scale and with more straightforward benchmarks, which are sufficiently covered in our experiments.
>
> **(2) Indirect evaluation**
>
> Recently, some research views that reasoning skill is learned directly from the dataset as a part of pattern memorization, rather than emerging as a mere intellectual invention [5,6].  From this view, if the training dataset contains more data that is similar to the targeted complex task benchmark, the model’s performance on that target benchmark would increase accordingly. This is because the model can utilize a larger number of direct templates to solve the task, in a manner of pattern memorization.
>
> From this concept, we mixed the targeted benchmark dataset (e.g., MMLU-pro) into the Dolma corpus and measured how much of this benchmark data is maintained through each filtering method. This simulates a scenario where the training dataset contains data similar to the target benchmark.  Therefore, if more benchmark data remain after filtering, we can assume there are higher likelihood that such data (i.e., similar to the targeted benchmark) is preserved in a larger-scale setting, leading to better benchmark performance. Although we can not directly evaluate MMLU-pro performance with a small LM (usually under 7B), this may offer an indirect way to estimate the potential impact of filters on complex task-solving capability.
>
> $\color{white}$
>
> ETable 2. The remaining portion of the MMLU-Pro dataset for each filtering method and threshold.
>
> | threshold (<) | ±0.2 | ±0.25 | ±0.3 | ±0.4 | ±0.45 |
> | --- | --- | --- | --- | --- | --- |
> | ppls-based | 0.222 | 0.292 | 0.436 | 0.740 | 0.987 |
> | prior-based | 0.510 | 0.629 | 0.743 | 0.931 | 0.990 |
> | random | 0.4 | 0.5 | 0.6 | 0.8 | 0.9 |
>
>
> $\color{white}$
>
> As a result, we find that the prior-based filter achieves significantly better recognition performance on MMLU-Pro data compared to PPL and random selection. Based on this observation, it is difficult to argue that a model trained with PPL-based filter would achieve a better score than a prior-based filter on MMLU-Pro.
>
> We analyze that this might be due to our method’s better preservation of “rare but valuable data,” as discussed in Appendix B.1. MMLU-Pro is known to contain a lot of logical statements, which likely constitute only a small portion of web corpora (e.g., Dolma). As prior-filter is more sensitive to the rare group (Appendix B.1), it might have a higher chance of capturing such data.
>
> $\color{white}$
>
> **#. Reviewer still might not be satisfied, but the scale demanded by the reviewers is far beyond what a typical research environment can support.**
>
> We have conducted the experiments on eight 25GB GPUs (A5000), and full runs (including ablations) took about 2 months of pure compute time (excluding other overhead). Also, only feasible model size for full-parameter tuning was 1.5B (not LoRA tuning). We guess this represents the average level of resources available to most researchers in academia, except for a small number of labs associated with large companies.  Reviewers requested training the larger model on a 100B-scale corpus, but this is physically impossible for us, or would require **about** **two years of computation.**
>
> Of course, we understand that there may always be such concern about the experiment scale.  Nevertheless, we believe our methodology is based on interesting and novel ideas, and thus worth sharing despite the limitations. To the best of our understanding, the academic community has followed an implicit protocol: when interesting theoretical attempts are made under constrained experimental environments in academia, companies with enough resources have often cross-validated these. We kindly ask the reviewers to take another look at the novel linguistic grounding and efficiency. Furthermore, we additionally provide a theoretical guarantee for our method’s noise-detection behavior, supporting that it works regardless of scales (Response to **HKmF[W1, W4]**).

---

> ### Author Response · Authors · 2025-11-23
> **Official Comment to Reviewer edXW <3>**
>
> **[W3, Q3]  Further discuss the filtering effectiveness for n-grams.**
>
> = This comment suggests that our method may have been misunderstood. **Our method operates on a principle that is opposite to that of n-gram approaches.** Therefore, our method is supposed to perform best in a 1-gram setting. We will clarify this point.
>
> $\color{white}$
>
> **Our method needs a clear separation between function and content words**
>
> In our method, frequency is used **only as a representation** for distinguishing function words and content words, and the mean and standard deviation measure whether the balance between these two categories falls within a normal range.
>
> Thus, our method requires a clear separation of function words from content words. However, with n-grams, function words are inevitably combined with adjacent content words, making it impossible to maintain a clean separation (e.g., "a word" "gram is"). In this case, it becomes impossible to measure the proportion of function words, causing the core mechanism of our method to collapse.
>
> $\color{white}$
>
> **Conversely, n-grams are useful when focusing only on content words**
>
> Opposite to our method, n-gram–based methods are typically used to reduce the influence of function words and to clarify the contextual nuances of content words,  and they indeed offer advantages in that purpose. For example, the meaning of the content word “*bank”* becomes much clearer when combined with adjacent words (e.g.,  “*river bank”* or “*money in bank”*). In such cases, n-grams are certainly more effective than 1-gram.
>
> However, our method does not consider the contextual meaning of content words. Instead, it focuses solely on the **occurrence ratio between content words and function words**, using this ratio to determine whether a paragraph exhibits a proper grammatical form.
>
> $\color{white}$
>
> **Context-agnostic property of prior-based filter**
>
> This **context-agnostic** property is also an advantage of our method. For example, in n-gram–based approaches, if the corpus contains many documents about “money,” texts from other contexts (e.g., “health”) may lie far from the central distribution. As a result, data from rare contexts with well-formed sentence structures may be filtered out. In contrast, our method considers only whether the grammatical structure is intact, regardless of the underlying context.
>
> $\color{white}$
>
> **#. DSIR is n-gram-based method.**
>
> Most importantly, one of the representative n-gram–based approaches is DSIR, which is one of our baselines. So, we compared our method with n-gram-based method, and ours outperforms DSIR (Table 2).
>
> ---
>
> $\color{white}$
>
> **[W2] In terms of writing, Chapter 2 includes excessive discussion of basic NLP concepts such as word frequency. I suggest that the authors reduce such background explanations or move them to the appendix.**
>
> = We are concerned that this request may have arisen from a misunderstanding of our method, seeing your other question (e.g., W3). Our method differs from conventional frequency-based approaches, and the background of this distinct usage is clearly explained in Chapter 2. As this distinct usage is our key contribution, this part cannot be omitted.
>
> §2.2 provides the foundation for how frequency serves as a representation of function and content words, and what properties this representation entails. Bringing the linguistic observation is the only way of justifying this concept, so it is an essential part.
>
> §2.1 provides the background needed to explain how our prior mean and prior standard deviation approximate the behavior of the existing PPL-based method. Since our method conceptually approximates a PPL-based filter (noted from line 195, and a more detailed theoretical explanation in response to **HKmF [W1,W4]**), it is necessary to clarify the role of each component of PPL.
>
> We will revise the final version to make this connection clearer.

---

> ### Author Response · Authors · 2025-11-23
> **Official Comment to Reviewer edXW <4>**
>
> **[W4, Q4] Filtering thresholds and filtering ratios.**
>
> ETable 3. Performance of large (1.5B) models on Dolma, under different threshold settings.
>
> | Threshold | Avg | World knowledge | Commonsense reasoning | Language understanding | Symbolic problem solving | Reading comprehension |
> | --- | --- | --- | --- | --- | --- | --- |
> | 50% | 9.20 | 9.53 | 11.27 | 10.31 | 11.13 | 3.79 |
> | 81% (elbow) | 8.79 | 10.04 | 7.67 | 8.97 | 13.04 | 4.21 |
>
> $\color{white}$
>
> = This topic is already explained in **§B.2**.
>
> (1) Our work conceptually builds on [1]. This work [1] experimented with various filtering ratios using PPL and reported that **50%** performs the best. And the key of our paper is to show that replacing PPL with a prior-based metric yields a more efficient alternative while maintaining comparable performance. So we maintained the optimal hyperparameters, including thresholds.
>
> (2) We also explored other thresholding strategies, “elbow”
>
> We also explored another thresholding method, “elbow”, where the threshold is set at the point where the prior metric changes sharply (beyond a certain gradient) on both sides. In the elbow method, a total of 81% of the data were selected. However, the performance was not better than simply using 50% threshold.
>
> (3) We are preparing for experiments with a smaller threshold (25%). If time permits, we will provide the results within the discussion period.
>
> (4) We use a fixed percentile as the threshold, and no other ratio is used as a hyperparameter.
>
> ---
>
> $\color{white}$
>
>
> **[Q1] Have the authors considered combining different methods to achieve better filtering performance?**
>
> =  Dolma [4] corpus itself already incorporates multiple filtering stages—including heuristic filtering, quality filtering, and even model-based filtering.  Our method achieves substantial performance gains on top of this, proving its compatibility when combined.
>
> < Filters applied in Dolma >
>
> - Heuristic rule : adopted from rules used in C4, Gopher.
> - Language filter (classifier-based) :  Using fastText to select English data.
> - Content filter (classifier-based) : Using fastText trained on the Jigsaw Toxic Comment dataset to filter out toxic documents
> - Deduplication (rule-based) : duplication regarding an exact match of URL, document, or paragraph.
>
> ---
>
> $\color{white}$
>
> **[Q5] Would different tokenizers significantly affect the filtering results? Moreover, can the filtering method remain effective across different tokenizers?**
>
> = #. Our method generalizes to most tokenizers, as most tokenizers (e.g., BPE, WordPiece, SentencePiece) have tendency to merge characters with high co-occurrence probability into a single token (e.g., “e” + “r” → “er”). This is because most tokenizers, either directly or approximately, follow the Minimum Description Length (MDL) [8] objective, which aims the best efficiency: to minimize both the vocabulary size and the length of representation. For example, the token *“er”* frequently co-occurs with various words (e.g., “cooler”, “teacher”), so assigning it as an independent token (e.g., “cool”, “teach”, “er”) is more efficient than assigning *“*cooler*”* and *“teacher”* as separate tokens, meaning it can cover wider range of expression with fewer vocab
>
> In short, in any tokenizers where content morphemes and function morphemes tend to be separated, our method is applicable.
>
>
> $\color{white}$
>
>
> #. Experiment
>
>
>
> ETable 4. Performance of large (1.5B) models on Dolma, under different tokenizers.
>
> | Tokenizer | Avg | World knowledge | Commonsense reasoning | Language understanding | Symbolic problem solving | Reading comprehension |
> | --- | --- | --- | --- | --- | --- | --- |
> | GPT2 | 9.20 | 9.53 | 11.27 | 10.31 | 11.13 | 3.79 |
> | LLaMA-3 | 9.39 | 9.54 | 11.16 | 10.78 | 11.86 | 3.64 |
> | T5-small | 8.11 | 8.59 | 7.43 | 7.95 | 12.32 | 4.22 |
>
>
> $\color{white}$
>
> = We clarify that our draft already included the comparison with two additional tokenizers (**§B.2, referred from §4.1, line 453**): LLaMA-3-8B (UTF-8-based, vocab size 128K), and T5-small
> (SentencePiece-based, vocab size 32K). GPT-2 tokenizer (mainly used in our paper) has a vocab size of 50K. All of these tokenizers tend to assign function morphemes as individual tokens in accordance with the MDL principle. However, we measure the performance differences among different granularities.
>
> The result is reported in ETable 4 (also Table 4 of our paper). All tokenizers showed the advantage of filtering, while T5 exhibited slightly lower performance gain. We propose two possible interpretations: (1) There may exist a threshold of optimal granularity, which lies between 32K and 50K. (2) Using the tokenizer paired with the model may offer stability.
>
> Nevertheless, as demonstrated, priors can be computed even with tokenizers mismatched to the model. If an optimal tokenizer exists, it can be freely adopted at any time.

---

> ### Author Response · Authors · 2025-11-23
> **Official Comment to Reviewer edXW <5>**
>
> Thank you very much for your time!
>
> [1] Z. Ankner et al., Perplexed by perplexity: Perplexity-based data pruning with small reference models, ICLR 2025
>
> [2]  Jeffrey Li et al., DataComp-LM: In search of the next generation of training sets for language models, NeurIPS DB 2025
>
> [3] Shizhe Diao et al.,   CLIMB: CLustering-based Iterative Data Mixture  Bootstrapping for Language Model Pre-training, NVIDIA ,2025
>
> [4] Luca Soldaini et al., Dolma: An Open Corpus of Three Trillion Tokens for Language Model Pretraining Research, ACL 2025
>
> [5] Zhaofeng Wu et al., Reasoning or Reciting? Exploring the Capabilities and Limitations of
> Language Models Through Counterfactual Tasks,  NAACL 2024
>
> [6] Iman Mirzadeh et al., GSM-Symbolic: Understanding the Limitations of Mathematical Reasoning in Large Language Models, ICLR 2025
>
> [7] Cleveland, Harlan. Information as a resource. *Futurist* 16.6 (1982): 34-39.
> [8] Hou, Jue, et al. Effects of sub-word segmentation on performance of transformer language models,  EMNLP 2023
>
> [9] Ruihang Li, et al., ScalingFilter: Assessing Data Quality through Inverse Utilization of Scaling Laws, ACL 2024
>
> [10] Guilherme Penedo, The FineWeb Datasets: Decanting the Web for the  Finest Text Data at Scale, NeurIPS DB 2024
>
> [11] Shayne Longpre et al., A Pretrainer’s Guide to Training Data: Measuring the Effects of Data Age, Domain Coverage, Quality, & Toxicity, NAACL 2024
>
> [12] S. M. Xie et al., Data selection for language models via importance resampling, NeurIPS 2023

---

> ### Comment · Reviewer_edXW · 2025-11-27
>
> Thank you for your efforts! I will raise my score.

---

> ### Author Response · Authors · 2025-11-28
> **Additional experiments**
>
> Thank you for raising the score. We provide additional experiments to address your remaining concerns.
>
> $ \color{white} dd $
>
> **#. Additional results on larger models.**
>
> EETable1. Overall average performance on benchmarks of the Qwen/Qwen2.5-3B model at each pre-training token duration (0.75B - 6B).
>
> | Token duration | 0.75B | 1.5B | 3B | 6B |
> | --- | --- | --- | --- | --- |
> | Prior-based | 2.32 | 3.42 | 6.36 | 11.35 |
> | PPL-based | 1.67 | 2.54 | 5.29 | 9.83 |
>
> $ \color{white} dd $
>
> EETable2. Results from training the Qwen/Qwen2.5-3B model for a 6B token duration (6B tokens × 1 repeat.
>
> |  | Avg | World knowledge | Commonsense reasoning | Language understanding | Symbolic problem solving | Reading comprehension |
> | --- | --- | --- | --- | --- | --- | --- |
> | Prior-based | 11.35 | 13.52 | 12.92 | 11.95 | 12.42 | 5.94 |
> | PPL-based | 9.83 | 11.12 | 13.56 | 8.59 | 9.59 | 6.31 |
>
> $ \color{white} dd $
>
> We provide additional experiments of the core baselines (prior- and PPL-based) on a larger model architecture, Qwen2.5-3B. We pretrained it from scratch for 6B token duration (6B x 1 repetition), and the results were consistent with those observed in the smaller-scale model (e.g., 1.5B). We will report the 12B-token duration results in the camera-ready version.
>
> The 3B model size is the **same as the largest model scale used in our core reference [1], which is last year’s ICLR paper**. It is also larger than the model sizes used in many recent studies on data filtering **[9,10,11,12,3]**. Therefore, our setup meets the standard level of rigor, and the results demonstrate that our method operates consistently across multiple scales. Also, the theoretical reason why our method yields performance gains regardless of scale is also well explained in **[W1]** and in response to **HKmF [W1]**.
>
> For this experiment, we rented four A100 GPUs (costing about \\$2k over two weeks, expected to total \\$4k by the end). We used a learning rate of 1.6e-4 following [1].
>
> $ \color{white} dd $
>
> ---
>
> **#. Additional experiments at different thresholds.**
> We additionally trained a 1.5B model for a 6B-token duration (i.e., same setting as experiment of §4.1) using a 25% threshold filter. Its performance was lower than the 50% threshold, likely because the 25% setting over-filtered not only noise but also meaningful information. Among the three thresholds we tested — 25%, 50%, and 81% — the 50% threshold yielded the best performance. This result is also consistent with the findings of [1], as we discussed in §B.2.
>
> $ \color{white} dd $
>
> EETable 3. Performance of large (1.5B) models on Dolma, under different threshold settings.
>
> | Threshold | Avg | World knowledge | Commonsense reasoning | Language understanding | Symbolic problem solving | Reading comprehension |
> | --- | --- | --- | --- | --- | --- | --- |
> | 25% | 6.21 | 7.03 | 7.06 | 5.99 | 7.36 | 3.57 |
> | 50% | 9.20 | 9.53 | 11.27 | 10.31 | 11.13 | 3.79 |
> | 81% (elbow) | 8.79 | 10.04 | 7.67 | 8.97 | 13.04 | 4.21 |
>
> $ \color{white} dd $
>
> **#. Request for more engagement in the discussion**
>
> We believe we have addressed all of the reviewer’s concerns. We would appreciate it if you could confirm whether there are any remaining issues, and if so, clearly indicate which points they refer to.

---

> > ### Author Response · Authors · 2025-12-03
> > **Additional experiments 2**
> >
> > #. Additional experiment on extended scale.
> >
> > EETable1. Overall average performance on benchmarks of the Qwen/Qwen2.5-3B model at each pre-training token duration (0.75B - 12B).
> >
> > | Token duration | 0.75B | 1.5B | 3B | 6B | 12B |
> > | --- | --- | --- | --- | --- | --- |
> > | Prior-based | 2.32 | 3.42 | 6.36 | 11.35 | 13.80 |
> > | PPL-based | 1.67 | 2.54 | 5.29 | 9.83 | 11.73 |
> >
> > EETable2. Results from training the Qwen/Qwen2.5-3B model for a 12B token duration (6B tokens × 2 repeat).
> >
> > |  | Avg | World knowledge | Commonsense reasoning | Language understanding | Symbolic problem solving | Reading comprehension |
> > | --- | --- | --- | --- | --- | --- | --- |
> > | Prior-based | 13.80 | 15.46 | 20.77 | 12.96 | 13.66 | 6.16 |
> > | PPL-based | 11.73 | 12.63 | 16.3 | 13.7 | 9.83 | 6.2 |
> >
> > Continuing from the last report, we present results from training a larger model (3B) at an expanded training scale (a total of 12B tokens, 6B tokens x 2 repeat), with core baselines (PPL- and prior-based). The superiority of our method remains consistent, demonstrating that the advantages of our approach hold even at larger scales.

---

### Official Review · Reviewer_HKmF · 2025-10-31

**Soundness:** 3
**Presentation:** 3
**Contribution:** 3
**Rating:** 6
**Confidence:** 2

**Summary:**

This paper proposes a text data filtering method based on word frequency prior to replace mainstream perplexity-based filtering strategies. Drawing upon statistical patterns of word frequency and lexical density in linguistics, the method identifies and filters anomalous documents deviating from the overall distribution by calculating the mean and standard deviation of the logarithm of word frequency within documents. Experimental results demonstrate that this method achieves performance superior to or equivalent to perplexity-based filtering across 20 downstream tasks, while reducing filtering time to approximately 0.1% of the original method, exhibiting remarkable computational efficiency. Furthermore, the method shows good applicability in both symbolic languages and multilingual corpora, without requiring pre-trained models or manually annotated data.

**Strengths:**

1. This method employs word frequency statistics for filtering, eliminating the need for training or invoking reference models, thereby reducing implementation complexity and resource dependency.

2. Across 20 downstream tasks encompassing world knowledge and common-sense reasoning, the model demonstrated superior average performance compared to baseline models.

3. This method is linguistically grounded in word frequency and lexical density, possessing an interpretable statistical foundation.

**Weaknesses:**

1. The paper should thoroughly discuss the advantages of PPL in capturing semantic coherence and contextual dependencies. Emphasizing only its computational cost and noise sensitivity issues may be misleading.

2. The author should consider incorporating additional lightweight or baseline metrics (such as character repetition rate and language model scores) for comparison to enhance the persuasiveness of the experimental results.

3. The authors should strengthen the discussion on the scaling relationship between computational efficiency and corpus size. Since experiments were conducted only at a 6B-word scale, they should demonstrate feasibility at larger scales.

4. Although the hypothesis of a priori approximate PPL is proposed, it lacks rigorous theoretical derivation or boundary condition analysis.

**Questions:**

1. The method uses the median as the central estimate for “normal documents.” For corpora with highly skewed distributions or mixed multimodal distributions (e.g., a blend of professional forums and news), is the median still a robust estimator? Have other robust central measures been considered?

2. The experiment demonstrated the effectiveness of processing mixed Chinese-English corpora. However, for variants within the same language (such as academic English versus colloquial English), word frequency distributions may differ. Can the method distinguish stylistic variations within the same language from genuine noise?

---

> ### Author Response · Authors · 2025-11-23
> **Official Comment to Reviewer HKmF <1>**
>
> Thank you very much for the sharp and insightful questions—they were truly helpful in clarifying our work, and addressing these questions was an enjoyable experience. We will make sure to include the discussions in the revision or the final version. Thank you very much for your time!
>
> ---
>
> $\color{white}$
>
> **[W1]  Advantages of PPL in capturing semantic coherence and contextual dependencies.**
>
> **[W4]  Theoretical derivation and boundary condition analysis of our method.**
>
> = We devoted §3.4.1 to analyzing what properties of a document PPL captures, and based on this analysis, Figure 3 illustrates which types of noise PPL is better at detecting compared to ours. Nonetheless, we provide a clearer and more detailed explanation here, and we will include it in the final paper.
>
> Through the following, we show that (1) under the linguistic observation presented in §2.2, the prior-based metric shares similar behavior with PPL in distinguishing noise. In particular, (2) we demonstrate that deviations from a central point can be interpreted as noise. (3) We also explain the boundary conditions under which this property does not hold.
>
> $\color{white}$
>
> ### ###. Theoretical guarantee of noise detection behavior
>
> **#.The components of PPL and the approximation via prior-based metric**
>
> As is well known, PPL consists of the following two components: the likelihood and the prior term.
>
> $\log PPL(d) \propto \sum_i \log p(x_{< i}|x_i) + \sum_i \log p(x_i)$
>
> And in §3.2, we introduced two prior-based metrics:
>
> (1) $\mu_d \propto \sum_i\log p(x_i)$
>
> (2) $\sigma_d =std[p(x_i)] \propto \sum_i[p(x_i) - e_d]^2 , \ e_d=E[p(x_j)]$
>
> **Prior term of PPL**
>
> As can be seen, the $\mu_d$ is identical to the prior term of PPL. Therefore, the prior-based filter partially approximates PPL.
>
> **Likelihood term of PPL**
>
> The $\sigma_d$ and likelihood share a similarity in that they reflect relationships among tokens rather than averaging the independent properties of individual tokens. However, it is difficult to formally prove a universal connection.
>
> However, under the following assumptions and scenarios, they **exhibit similar behavior in filtering noisy data**.
>
> $\color{white}$
>
> **Assumptions:**
>
> We assume very simplified settings of linguistic observation in §2.2.
>
> (1) Suppose tokens consist of only two categories, *F* and *C*, where *F* represents function words and *C* represents content words.
> (2) Within any well-formed document, these two groups must follow a fixed ratio $r^*$ (e.g., F:C=4:1).
>
> (3) Actually, data $d$ has a fixed length (e.g., 5), and has an observed ratio $r$. And $r$ has only two possible values:  $r^*$(well-formed) or $\hat{r}$ (noise).
>
> Example of  $r^*$ : FFFCF, FFCFF
>
> Example of  $\hat{r}$ : FFFFF, FFCCF
>
> (4) If the data of $r^* $ takes the big proportion in the corpus, both the empirical average of ratio and the trained model $p$ converge near to $r^*$.
>
>  (1)–(3) are simplified versions of the linguistic observation in §2.2, and (4) represents the minimal condition required for our method to operate. (Experiment in §3.4.2 shows the minimal proportion is around 10%)
>
> $\color{white}$
>
> **Property of conditional likelihood**
>
> In this simplified setting, conditional likelihood is determined solely by $r$  (e.g.,    $p(\text{else} \mid C) \ll p(\text{FFFF} \mid C)\approx 1$ ).
>
> Therefore, the distribution of $l_d =\sum_i \log p(x_{\neq i}|x_i)$ places data with r* near to 1, while placing $\hat{r}$ near the left tail. And since  $r^*$ dominates the corpus (assumption 4), the overall distribution of $l_d$ becomes right-skewed.
>
> Therefore, in the distribution of $l_d$, data with ratio $\hat{r}$ can be identified through the left tails that deviate far from the representative value (e.g., median) of the distribution.
>
> **Property of $\sigma_d$**
>
> For the $\sigma_d$, the computation is straightforward. Given a fixed occurrence probabilities (e.g., p(C) = 1/5 and p(F) = 4/5), the $\sigma_d$ are distributed around specific value (e.g.,  $E[\sigma_d]=0.26$)  where the ratio of data is $r^*$. And the samples that deviate far from this point can be distinguished as the one with $\hat{r}$.
>
> Therefore, both the likelihood term and the $\sigma_d$ exhibit similar behavior in identifying $\hat{r}$.
>
> **Property of $\mu_d$**
>
> Under the FC assumption, we further explain the role of the prior term of PPL ($\mu_d$) in distinguishing $\hat{r}$. Suppose that F and C have fixed occurrence probabilities (e.g.,  $p(F)=4/5, \ p(C)=1/5$)  and appear within each document according to those probabilities. Then, we can observe that the average of their occurrence probabilities tends to concentrate around a specific value (e.g., $E(\mu_d)=0.68$).
>
> If the observed ratio deviates from this central value, it will move further away from the central value, allowing us to identify $\hat{r}$  accordingly.

---

> ### Author Response · Authors · 2025-11-23
> **Official Comment to Reviewer HKmF <2>**
>
> **In conclusion, under assumptions 1–4, we can guarantee two properties of prior-based metrics.**
>
> (1) the prior-based metrics ($\mu_d, \sigma_d$) align with all components of PPL in identifying $\hat{r}$. So, we can say the prior-based filter “weakly aligns” with PPL-based filter, as we mentioned in line:264.
>
> (2) the prior-based metric of normal data with valid structures ($r^*$) converges around a single central point, allowing outliers ($\hat{r}$) to be distinguished by their distance from this center.
>
> $\color{white}$
>
> **Discrepancy**
>
> However, such an approximation required simplified assumptions. Consequently, when these assumptions are violated in a real-world data, a discrepancy arises.
>
> The most significant violation case we can identify is the following: even when a document follows the ratio $r^*$, it may still contain no meaningful information.
>
> This corresponds to cases such as:
>
> (1) Text with an appropriate balance ($r^*$) of function and content words, yet are semantically nonsensical. **This case is illustrated with Figure 3(a)**.
>
> (2) The data are not noise, nor semantically nonsensical; rather, they are either too difficult or too trivial to be useful for the model. (In this scenario, the main LLM should compute PPL directly, so this case is not applicable to our PPL baseline either.)
>
> In such cases, PPL will play a clearer role, and the prior-based filter will not be able to handle this, and we do not claim this. Prior filter is designed to identify only the noisy data ($\hat{r}$), as we emphasized in the title and §1.
>
> However, we claim the following:
>
> (1) The prior can at least provide a clearer signal in identifying $\hat{r}$, as model-predicted likelihood can often be unreliable.
>
> (2) In the corpus with a very high proportion of noise (e.g., web crawled corpus), identifying $\hat{r}$ may be more impactful than deliberate reordering among $r^*$ data.
>
> (3) Still, compatible with PPL-based filter.

---

> ### Author Response · Authors · 2025-11-23
> **Official Comment to Reviewer HKmF <3>**
>
> **[W2] Additional lightweight or baseline metrics**
>
> ETable 2. Performance of large (1.5B) models on Dolma.
>
> | Type, source |  | Avg | World knowledge | Commonsense reasoning | Language understanding | Symbolic problem solving | Reading comprehension |
> | --- | --- | --- | --- | --- | --- | --- | --- |
> | rule-based | vanilla Dolma | 5.78 | 5.52 | 0.44 | 6.14 | 13.22 | 3.59 |
> | classifier | FastText | 7.09 | 6.71 | 6.11 | 6.89 | 11.93 | 3.82 |
> | n-gram, classifier | DSIR | 7.56 | 7.03 | 6.84 | 7.31 | 12.67 | 3.97 |
> | model score | PPL-based | 8.22 | 9.98 | 11.91 | 7.34 | 7.91 | 3.96 |
> | prior | Prior-based | 9.21 | 9.53 | 11.27 | 10.31 | 11.13 | 3.79 |
>
> $\color{white}$
>
> We acknowledge this concern. However, as explained in the Related Work section (§A, line 694), we selected representative methods for each category and compared them. While it would be ideal to include as many baselines as possible, we only compared representative ones due to the substantial computational resources of pretraining.
>
> We explain which aspects each baseline represents.
>
> (1) vanilla Dolma (noted as “no-filter” in the paper)
>
> Dolma [3] corpus itself already incorporates multiple filtering stages—including heuristic filtering, quality filtering, and even model-based filtering.
>
> - Heuristic rule: adopted from rules used in C4, Gopher.
> - Language filter (classifier-based) :  Using fastText to select English data.
> - Content filter (classifier-based) : Using fastText trained on the Jigsaw Toxic Comment dataset to filter out toxic documents
> - Deduplication (rule-based) : duplication regarding an exact match of URL, document, or paragraph.
>
> Our method achieves substantial performance gains on top of this. Moreover, these filters are usually computationally lightweight and are often applied already during the web-crawling stage [2,3]. Therefore, they are compatible with our method by default.
>
> (2) DSIR
>
> DSIR operates on n-gram embedding vectors trained on a curated dataset. This work [4] also compares a classifier-based embedding option (FastText). So this baseline covers both n-gram and classifier-based methods.
>
> (3) PPL-based
>
> The [2] paper compares a wide range of methods, and demonstrates that PPL-based approach outperforms well-known model-based methods (AskLLM, classifier on BGE features, PageRank, SemDedup) and heuristic filters, or is comparable to FastText and top-k average logits.  Therefore, we selected the PPL-based method as the baseline representing other classifier-based methods.
>
> (4) FastText
>
> To directly compare against classifier-based approaches, we additionally provide the comparison with FastText (trained under the same dataset as DSIR). Its performance was similar but slightly lower than DSIR, which is consistent with the findings reported in [4]. The limitations of curation-based methods are also discussed in A.1.3.
>
> #. Conclusion
>
> In summary, our experiments cover various types of baselines, including heuristic rule-based, n-gram-based, and classifier-based methods. In addition, we provide an extra classifier-based baseline experiment for the sake of clarity.
>
> ---
>
> $\color{white}$
>
> **[W3] Discussion on the scaling relationship between computational efficiency and corpus size.**
>
> = (1) Filter efficiency on an extended scale.
>
>  If the question is about the filtering speed on large-scale corpora, we can say that the processing time of our method generally increases linearly with the data size.
>
> Our method took about 10 CPU minutes to process a 6B-token corpus (5 minutes for calculating token priors and 5 minutes for inference). The inference part increases linearly with the data size, whereas the token prior calculation involves sampling and therefore can be performed on a smaller subset.
>
> For example, in the 6B dataset, the outliers identified using a 1% subset were almost identical to those identified using the full 100% dataset (§3.4.3). In the case of the 12B dataset, this property emerged even with a 0.5% subset. Consequently, processing the 12B dataset took about 15 minutes (5 minutes for calculating token priors and 10 minutes for inference). We expect the same principle to hold for larger datasets.
>
> $\color{white}$
>
> ETable1. The same experimental setting as in Figure 6 of the paper, conducted on a 12B-size dataset.
> When the token prior is computed using a *b%* subset of Dolma, we count the proportion of outliers overlapping with those from *b = 100.*
>
> | Sample ratio (b) | Outlier overlap |
> | --- | --- |
> | 0.1% | 0.7709 |
> | 0.5% | 0.9719 |
> | 1% | 0.9773 |
> | 10% | 0.9798 |
>
> $\color{white}$
>
> (2) If your question is a request for results at a larger scale, we also acknowledge this issue and discussed it with Reviewer **533u [W3-1].** We apologize for not including the full details here, as we did not want to occupy excessive space if this was not your intended concern.

---

> ### Author Response · Authors · 2025-11-23
> **Official Comment to Reviewer HKmF <4>**
>
> **[Q1]  For corpora with highly skewed distributions or mixed multimodal distributions (e.g., a blend of professional forums and news), is the median still a robust estimator? Have other robust central measures been considered?**
>
> **[Q1-1] Skewed distribution**
>
> =A skewed distribution would refer to a case where the volumes of the left and right tails of the distribution are significantly different. However, we observed that distribution of prior-based metrics usually are not (Figure 2), which is why we chose to use the median. Since Dolma is inherently a mixed dataset (news, books, code), Figure 2 shows that this pattern remains consistent even in the mixed setting.
>
> We also experimented with the “elbow” method (§B, Table 4 “elbow”), where the threshold is set at the point where the prior metric changes sharply (beyond a certain gradient) on both sides. This method can handle cases where the lengths of the two tails are different. However, the performance was not better than using a simple median (line 942). Nevertheless, this approach can be applied when faced with a more skewed distribution.
>
> $\color{white}$
>
> **[Q1-2] Mixed multimodal distributions**
>
> =  **(1)** Prior-based filter only measures the grammatical structure via the  function-content word ratio. And texts in one specific language tend to **share the same grammatical structure, regardless of their style or domain** (e.g., news, forum). Thus, prior-filter will generalize to each other.
>
> Dolma is such an example. Although Dolma consists of various types of data (e.g., news, books, symbolic scripts), our method has been shown to perform well across these domains.
>
> **(2)** Std metric ($\sigma_d$) measures the distance between token priors, thus the **distance itself may not vary much among other types of language.**
>
> Let’s bring the “FC” assumption from **[W1,4]** to here. Let's assume there are heterogeneous language groups A and B, and each has its function-content words pair, and each group follows the assumption of W1,4.  $\{F,C\} \in A, \ \{R,T\} \in B$.
>
> According to the linguistic axiom, we can assume that distinct linguistic system (A, B) shares a similar function-content ratio (e.g., 1:4). Therefore, the central value of std metric for both languages is the same (e.g., $E[\sigma^A]=E[\sigma^B]=0.26$). Thus, $\sigma_d$ for A and B share the same central value, even though they are heterogeneous.
>
> **(3)** However, in practice, if the amount of B is significantly smaller than A, then there can be an imbalance of the function-content ratio.  For example, p(F)=0.4, p(C)=0.1, p(R)=0.0004, p(T)=0.0001. In this case, the central value of the distribution of $σ^A$ and $σ^B$ will diverge. We suggest two points:
>
> **(3-1)** **We tested an extreme case** of this in §3.4.2 with the Chinese-English set, to answer the question “When two entirely different languages are mixed together, to what extent can a single median encompass both languages?” Our experiments indicate that if a minority group is mixed only over 10%, most of the minority language is still considered an inlier when using a single median.
>
> **(3-2)** If this is still insufficient, **we can compute the token prior by mixing in a corpus** that shares attributes with the target corpus. For example, if we mix in more data that resembles corpus B and then recalculate the token prior, we might obtain token priors such as p(F)=0.4, p(C)=0.1, p(R)=0.4, and p(C)=0.1. In this case, we can bring corpora A and B to the same median value again, even though the actual proportion of B is very small.
>
> This approach corresponds to the method described in §B.1.1 (2) of our paper. For example, when we compute the token prior using a setting where GitHub data are added to Dolma, the resulting token prior enables better identification of GitHub-like data (e.g., symbolic scripts) within Dolma.
>
> ---
>
> $\color{white}$
>
> **[Q2] Can the method distinguish stylistic variations within the same language from genuine noise?**
>
> = This seems to overlap with [Q1-2], as stylistic variation may be interpreted as a multimodal distribution. As addressed in Q1-2(1), the distribution of the prior metric is shared within the same grammatical group. Moreover, since the English–Chinese pair represents an extreme case of linguistic difference, groups with smaller differences would require an even smaller portion to share the same median (less than 10%).
>
> $\color{white}$
>
> ---
>
> Thank you very much for your time!
>
> [1] Z. Ankner et al., Perplexed by perplexity: Perplexity-based data pruning with small reference models, ICLR 2025
>
> [2] Jeffrey Li et al., DataComp-LM: In search of the next generation of training sets for language models, NeurIPS DB 2025
>
> [3] Luca Soldaini et al., Dolma: An Open Corpus of Three Trillion Tokens for Language Model Pretraining Research, ACL 2025
>
> [4] Sang Michael Xie et al., Data Selection for Language Models via Importance Resampling, NeurIPS 2023

---

> > ### Comment · Reviewer_HKmF · 2025-11-28
> >
> > Thanks for detailed response, I will keep my recommendation score.

---

### Official Review · Reviewer_533u · 2025-11-01

**Soundness:** 2
**Presentation:** 2
**Contribution:** 2
**Rating:** 4
**Confidence:** 3

**Summary:**

This paper proposes a prior-based filter that estimates a token's context free probability from corpus term frequencies (priors). For each document, it computes the mean and standard of token priors and drops outliers far from corpus medians, motivated by (i) frequency $\approx$ function/content role and (ii) well-formed text having a stable lexical density. This method approximates PPL filtering while being $~1000\cdot$ cheaper (no model inference) and often outperforms PPL-based and DSIR baseline across  nearly 20 downstream tasks. The code is reproducible.

**Strengths:**

1. Practical and speedy. This method requires only frequency counts and reported wall-clock is orders of magnitude lower than PPL-based pipelines.
2. Clear, language agnostic intuition: This paper links token frequency to function/content roles; uses document-level $\mu, \sigma$ to capture lexical density.
3. It covers 2 model sizes with 20 tasks. Extensive experiments.

**Weaknesses:**

1. Frequency-driven criteria can over-filter rare but valuable content (named entities, dialects, minority languages). This paper discusses some multi-lingual behavior in Section 3.4.2 but they use Chinese and English as an example. These two languages are very common in web. Broader language settings might be needed.
2. Baseline scope might be limited. Comparisons focus on PPL and DSIR; other lightweight filters (classifier-based, memorization/dup measures, n-gram LM perplexity, quality/explicit toxicity heuristics) are not covered.
3. Scaling to large models: The authors report results for 137M-1.5B GPT-2 like models. It's unclear whether the advantage persists at 7B-70B scales or with instruction-tuning datasets.

**Questions:**

Can the authors add more baselines to compare and see whether other lightweights methods have shorter time and also good performance? For instance, baselines listed in related work?

---

> ### Author Response · Authors · 2025-11-23
> **Official Comment to Reviewer 533u <1>**
>
> Thanks so much for participating in the review process. While many of the questions are constructive, some appear to stem from misunderstandings of our method **[W1]** or from not fully reading parts that are already sufficiently explained in the paper **[W1, W2, Q1]**. We kindly ask that you read the rebuttal and our paper carefully. All other meaningful discussions will be incorporated into the final version.
>
> ---
>
> $\color{white}$
>
> **[W1-1] Frequency-driven criteria can over-filter rare but valuable content (named entities, dialects, minority languages)**
>
> = Your statement “Frequency-driven criteria can over-filter rare but valuable content” seems to partly stem from a misunderstanding of our method.
>
> **#. Misunderstanding of our method.**
>
> Although our method is based on frequency, this does not mean that we simply filter out or penalize contents with low frequency. Frequency was used as a representation of how close a token is to function words or content words. And **we focus solely on the balance** between function words and content words within the data (the balance is reflected in $\mu,\sigma$ - Theoretical explanation in response to **HKmF [W1]**). Thus, text with extremely rare terms is well preserved if it is part of a regular function-content ratio (as already clarified in  Appendix B.1.2). From this perspective, the statement “our method penalizes documents with rare token patterns” stems from a misunderstanding.
>
> $\color{white}$
>
> **#. General over-filtering problem**
>
> Nevertheless, loss of rare but valuable data is an inherent cost paid by all filtering methods, and reducing this cost remains one of the goals in this field. About this issue, we already **very thoroughly discussed in** § **B.1 (referred from** §**4, line 452).**
>
> $\color{white}$
>
> **For your convenience, we will briefly repeat here.**
>
>  The aspect you mentioned has been partially measured through benchmarks, where our baselines demonstrate relative strengths. For example, the scenario you described can be observed in the performance on symbolic task benchmarks within Dolma. Dolma primarily consists of web data ( Common Crawl 74.6%, Table1), but contains a small amount of code and math scripts (Stack 13.4%), which fits the description of “rare and valuable data”.
>
> In Table 2, PPL-based models underperform compared to no-filter on these symbolic tasks, revealing the trade-off problem of the filtering method. However, our method achieves higher performance than the PPL-based approach, indicating its superior ability to leverage such high-value minority data.
>
> **Nevertheless, to address your concern, we introduce two additional methods for a better trade-off.**
>
> (1) Using only stds:
>
> While the mean reflects the average frequency of tokens, the stds calculate the variation among token frequencies. Therefore, it is more suitable for identifying well-formed text written in low-frequency languages. Empirically, the stds-only filter outperforms both the baselines overall and the stds+mean filter on symbolic tasks. (ETable1)
>
> (2) Calculate prior on blended corpus:
>
> Our second strategy is to incorporate target-domain data when estimating token frequencies, thereby assigning higher prior probabilities to domain-specific terms and preventing their exclusion during filtering. In practice, we mixed Pile-GitHub data with Dolma in equal proportion for prior computation, which needed only an additional 10 minutes of processing.
>
> As a result, overall performance improved beyond the original method, while symbolic task performance also increased (ETable1).
>
> $\color{white}$
>
> ETable 1:  Performance of 1.5B model with Dolma. The *Prior (Dolma + GitHub)* uses both std and mean
>
> |  | Avg | World knowledge | Commonsense reasoning | Language understanding | Symbolic task | Reading comprehension |
> | --- | --- | --- | --- | --- | --- | --- |
> | no-filter | 5.78 | 5.52 | 0.44 | 6.14 | 13.22 | 3.59 |
> | PPL-based | 8.22 | 9.98 | 11.91 | 7.34 | 7.91 | 3.96 |
> | Prior (stds + mean) | 9.20 | 9.53 | 11.27 | 10.31 | 11.13 | 3.79 |
> | Prior  (stds) | 8.70 | 7.28 | 10.57 | 9.34 | 12.40 | 3.89 |
> | Prior (dolma + github) | 9.48 | 11.57 | 10.83 | 8.97 | 12.27 | 3.78 |

---

> ### Author Response · Authors · 2025-11-23
> **Official Comment to Reviewer 533u <2>**
>
> **#. What if rare terminology is used? (e.g., technical paper)**
>
> We also already validated this scenario in a controlled setting in §B.1.2.
>
> We first consider a scenario where a document with a general structure contains extremely rare terminology. To simulate this, we sampled 1,000 data points (each  512-token length) from the Dolma dataset within the central ±15% range of $\mu$ and gradually injected rare terms into them. Rare terminology was generated by concatenating two tokens ranked in the bottom 10% of the prior distribution (e.g., “prosecromeda”, combining two tokens “prosec” and “romeda”). We inserted $n$ of these terms into each text and measured the percentage of these texts classified as outliers ( ±25% threshold). The result is as follows.
>
> $\color{white}$
>
> ETable 2: $n$ is the number of injected terminology. Inliers is the rate of texts remaining in 25%~75% boundary.
>
> | n | n * token length | inliers (25%~75%) |
> | --- | --- | --- |
> | 1 | 2 | 1.0 |
> | 6 | 12 | 1.0 |
> | 7 | 14 | 0.98 |
> | 8 | 16 | 0.91 |
> | 9 | 18 | 0.67 |
>
> $\color{white}$
>
> Up to 7 insertions of rare terms—amounting to 14 tokens (2.7%) within a 512-token block—were required before around 10% of texts were filtered out.
>
>  **To assess whether seven occurrences are a reasonable upper bound,** we examined the typical frequency of topic-specific terminology in real-world text. Specifically, we sampled 10k Wikipedia articles, segmented them into 512-token blocks, normalized all text to lowercase, and counted the occurrences of the article title (i.e., subject entity) within each block.
>
> On average, the title appeared 1.09 times per block, corresponding to an average token length of 3.44. This suggests that even documents intended to explain a given concept are far from being dominated by that terminology. Taken together, these findings indicate that the prior-based filter exhibits strong robustness when handling scenarios involving rare terminology.

---

> ### Author Response · Authors · 2025-11-23
> **Official Comment to Reviewer 533u <3>**
>
> **[W1-2] This paper discusses some multi-lingual behavior in Section 3.4.2 but they use Chinese and English as an example. These two languages are very common in web.**
>
> = (1) First, we clarify that the experiment in §3.4.2 already established a controlled setting to observe how much rare-language data are filtered out. Based on English data, we mix in Chinese at various ratios (1%, 5%, 10%, etc.). Therefore, the actual proportion of Chinese on the internet does not affect this experiment. According to the results, even when rare data account for only 10% of the mixture, the prior-based filter recognizes most of them as inliers.
>
> (2) Next, to further address your concern, we also conducted the same experiment with Turkish, which has a relatively small presence on the internet, and obtained similar results. This result confirms that our method’s multilingual behavior generalizes to other rare languages as well.
>
> $\color{white}$
>
> Etable 3.  The same setting as in Figure 5 of the paper. Proportion of Turkish data classified as outliers, after mixing Turkish and English data at a ratio of a : 100. Outliers are the
> top and bottom 5% of µd.
>
> | Size of turkish data compared to En (a) | Proportion of outliers (10%) |
> | --- | --- |
> | 1 | 0.5036 |
> | 5 | 0.2778 |
> | 10 | 0.2448 |
> | 20 | 0.1984 |
> | 100 | 0.1337 |
>
>
> ---
>
> $\color{white}$
>
> **[W2, Q1] Baseline scope might be limited.**
>
>
>
> ETable 4. Performance of large (1.5B) models on Dolma.
>
> | Type, source |  | Avg | World knowledge | Commonsense reasoning | Language understanding | Symbolic problem solving | Reading comprehension |
> | --- | --- | --- | --- | --- | --- | --- | --- |
> | rule-based | vanilla Dolma | 5.78 | 5.52 | 0.44 | 6.14 | 13.22 | 3.59 |
> | classifier | FastText | 7.09 | 6.71 | 6.11 | 6.89 | 11.93 | 3.82 |
> | n-gram, classifier | DSIR | 7.56 | 7.03 | 6.84 | 7.31 | 12.67 | 3.97 |
> | model score | PPL-based | 8.22 | 9.98 | 11.91 | 7.34 | 7.91 | 3.96 |
> | prior | Prior-based | 9.21 | 9.53 | 11.27 | 10.31 | 11.13 | 3.79 |
>
> $\color{white}$
>
> = We acknowledge this concern. However, as explained in the Related Work section (§A, line 694), we selected representative methods for each category and compared them. While it would be ideal to include as many baselines as possible, we only compared representative ones due to the substantial computational resources of pretraining.
>
> We explain which aspects each baseline represents.
>
> (1) vanilla Dolma (noted as “no-filter” in the paper)
>
> Dolma [8] corpus itself already incorporates multiple filtering stages—including heuristic filtering, quality filtering, and even model-based filtering.
>
> - Heuristic rule : adopted from rules used in C4, Gopher.
> - Language filter (classifier-based) :  Using fastText to select English data.
> - Content filter (classifier-based) : Using fastText trained on the Jigsaw Toxic Comment dataset to filter out toxic documents
> - Deduplication (rule-based) : duplication regarding an exact match of URL, document, or paragraph.
>
> Our method achieves substantial performance gains on top of this. Moreover, these filters are usually computationally lightweight and are often applied already during the web-crawling stage [2]. Therefore, they are compatible with our method by default.
>
> (2) DSIR
>
> DSIR operates on n-gram embedding vectors trained on a curated dataset. This work [4] also compares a classifier-based embedding option (FastText). So this baseline covers both n-gram and classifier-based methods.
>
> (3) PPL-based
>
> The [2] paper compares a wide range of methods, and demonstrates that PPL-based approach outperforms well-known model-based methods (AskLLM, classifier on BGE features, PageRank, SemDedup) and heuristic filters, or is comparable to FastText and top-k average logits.  Therefore, we selected the PPL-based method as the baseline representing other classifier-based methods.
>
> (4) FastText
>
> To directly compare against classifier-based approaches, we additionally provide the comparison with FastText (trained under the same dataset as DSIR). Its performance was similar but slightly lower than DSIR, which is consistent with the findings reported in [4]. The limitations of curation-based methods are also discussed in A.1.3.
>
> The SCDP method mentioned in the related work section was originally designed for finetuning, and could not be applied to billion-scale datasets. Upon examining the code, we found that it constructs a tensor of “dataset size × dataset size” when computing, which likely caused the limitation.
>
> #. Conclusion
>
> In summary, our experiments cover various types of baselines, including heuristic rule-based, n-gram-based, and classifier-based methods. In addition, we provide an extra classifier-based baseline experiment for the sake of clarity.

---

> ### Author Response · Authors · 2025-11-23
> **Official Comment to Reviewer 533u <4>**
>
> **[W3-1] Scaling to large models**
>
>
> ETable5.  Performance of models on Dolma. 12B token duration (1.5B model)
>
> |  | Avg | World knowledge | Commonsense reasoning | Language understanding | Symbolic problem solving | Reading comprehension |
> | --- | --- | --- | --- | --- | --- | --- |
> | PPL-based | 9.54 | 12.01 | 13.21 | 9.39 | 8.54 | 4.57 |
> | Prior-based | 10.51 | 11.43 | 12.92 | 11.21 | 12.73 | 4.28 |
>
> $\color{white}$
>
> = Due to resource constraints (we are using eight 25 GB GPUs), we were unable to increase the model size. Instead, we extended our experiments using a larger dataset. We additionally trained a checkpoint of 1.5B on an additional subset of Dolma, to be a total of 12B token duration (2 times repetition of 6B). The results were largely consistent with the previous one (Table 2), demonstrating the stable performance of our method.
>
> Below, we explain that (1) our model size maintains experimental rigor when viewed from previous representative studies, and (2) experiments with small-sized models are sufficient for evaluating filtering methods.
>
> $\color{white}$
>
> **#. Justification of our scale, based on previous works**
>
> Many representative works on data filtering employ relatively small models around 1.5B parameters, assuming that such scale is sufficient for reliable evaluation. For example, [1] uses 1B and 2B models, while [3] employs 350M and 1B models. In addition, recent representative works in this topic [9, 10, 11, 12] all conduct experiments with models smaller than 2B. This contrasts with studies focusing on model architectures (e.g., LLaMA, DeepSeek) that compete in complex reasoning performance, which typically require pretraining of large-scale models (e.g., 7B and 70B). A detailed rationale of why small model settings can still maintain experimental rigor in data filtering, is provided below.
>
> $\color{white}$
>
> **#. Noise filtering can be evaluated at a small model**
>
> Current demands for large-model experiments mostly come from the need for validating “complex reasoning skill”, because such a capability is known to usually emerge in larger models. As model size increases, models can learn more complex patterns, or models become capable of multi-step reasoning [13], so the complex reasoning ability emerges. If the goal of work is to demonstrate such inherent capabilities of model architecture, it would indeed make sense to compare models at a larger size, like works on foundational architecture (e.g., LLaMA, DeepSeek).
>
> However, our method is designed to filter noise, and its effectiveness can be evaluated at a modest size.  If we can assume noisy data as “documents that do not contain information”, what we need is simply to measure how much information the model has acquired from the dataset. This can be evaluated through straightforward tasks (e.g., world knowledge, commonsense reasoning, language understanding), which do not require multi-step reasoning or very complex pattern recognition. Thus, these tasks are known to be handled with modest-sized models.
>
> $\color{white}$
>
> **#. Reviewers still might not be satisfied, but the scale demanded by the reviewers is far beyond what a typical research environment can support.**
>
> We have conducted the experiments on eight 25GB GPUs (A5000),  and full runs (including ablations) took about 2 months of pure compute time (excluding other overhead). **In our resources, the maximum feasible model size for full-parameter tuning was 1.5B (not LoRA tuning)**. And we guess this represents the average level of resources available to most researchers in academia, except for a small number of labs associated with large companies.
>
> Of course, we understand that there may always be such concerns about the model scale.  Nevertheless, we believe our methodology is based on interesting and novel ideas, and thus worth sharing despite the limitations. To the best of our understanding, the academic community has followed an implicit protocol: when interesting theoretical attempts are made under constrained experimental environments in academia, companies with enough resources have often cross-validated these. We kindly ask the reviewers to take another look at the novel linguistic grounding and efficiency.  Furthermore, we additionally provide a theoretical guarantee for our method’s noise-detection behavior, supporting that it works regardless of scales (Response to **HKmF[W1,W4]**).
>
> ---
>
> $\color{white}$
>
> **[W3-2]  instruction-tuning datasets.**
>
> Our method is designed to filter noisy text data with broken sentence structures. Therefore, it provides the greatest value when applied to web corpora, because such corpora are (1) inherently noisy and (2) extremely large, making efficiency crucial. In contrast, instruction-tuning data are typically noise-free and relatively small in scale, where extreme efficiency is not a primary concern.

---

> ### Author Response · Authors · 2025-11-23
> **Official Comment to Reviewer 533u <5>**
>
> Thank you very much for your time!
>
> [1] Z. Ankner et al., Perplexed by perplexity: Perplexity-based data pruning with small reference models, ICLR 2025
>
> [2]  Jeffrey Li et al., DataComp-LM: In search of the next generation of training sets for language models, NeurIPS DB 2025
>
> [3] Shizhe Diao et al.,   CLIMB: CLustering-based Iterative Data Mixture  Bootstrapping for Language Model Pre-training, NVIDIA ,2025
>
> [4] S. M. Xie et al., Data selection for language models via importance resampling, NeurIPS 2023
>
> [5] Zhaofeng Wu et al., Reasoning or Reciting? Exploring the Capabilities and Limitations of
> Language Models Through Counterfactual Tasks,  NAACL 2024
>
> [6] Iman Mirzadeh et al., GSM-Symbolic: Understanding the Limitations of Mathematical Reasoning in Large Language Models, ICLR 2025
>
> [7] Cleveland, Harlan. "Information as a resource." *Futurist* 16.6 (1982): 34-39.
>
> [8] Luca Soldaini et al., Dolma: An Open Corpus of Three Trillion Tokens for Language Model Pretraining Research, ACL 2025
>
> [9] Ruihang Li, et al., ScalingFilter: Assessing Data Quality through Inverse Utilization of Scaling Laws, ACL 2024
>
> [10] Guilherme Penedo, The FineWeb Datasets: Decanting the Web for the  Finest Text Data at Scale, NeurIPS DB 2024
>
> [11] Shayne Longpre et al., A Pretrainer’s Guide to Training Data: Measuring the Effects of Data Age, Domain Coverage, Quality, & Toxicity, NAACL 2024
>
> [12] Cleveland, Harlan. Information as a resource. Futurist 16.6 (1982): 34-39.
>
> [13] Jason Wei et al.,, Chain-of-Thought Prompting Elicits Reasoning in Large Language Models. NeurIPS 2022

---

> ### Author Response · Authors · 2025-11-28
> **Additional experiment**
>
> We provide additional experiments to address your remaining concerns.
>
> $ \color{white}dd $
>
> **#. Additional results on larger-scale models.**
>
> $ \color{white}dd $
>
> EETable1. Overall average performance on benchmarks of the Qwen/Qwen2.5-3B model at each pre-training token duration (0.75B - 6B).
>
> | Token duration | 0.75B | 1.5B | 3B | 6B |
> | --- | --- | --- | --- | --- |
> | Prior-based | 2.32 | 3.42 | 6.36 | 11.35 |
> | PPL-based | 1.67 | 2.54 | 5.29 | 9.83 |
>
> $ \color{white}dd $
>
> EETable2. Results from training the Qwen/Qwen2.5-3B model for a 6B token duration (6B tokens × 1 repeat.
>
> |  | Avg | World knowledge | Commonsense reasoning | Language understanding | Symbolic problem solving | Reading comprehension |
> | --- | --- | --- | --- | --- | --- | --- |
> | Prior-based | 11.35 | 13.52 | 12.92 | 11.95 | 12.42 | 5.94 |
> | PPL-based | 9.83 | 11.12 | 13.56 | 8.59 | 9.59 | 6.31 |
>
> $ \color{white}dd $
>
> We provide additional experiments of the core baselines (prior- and PPL-based) on a larger model architecture, Qwen2.5-3B. We pretrained it from scratch for 6B token duration (6B x 1 repetition), and the results were consistent with those observed in the smaller-scale model (e.g., 1.5B). We will report the 12B-token duration results in the camera-ready version.
>
> The 3B model size is the **same as the largest model scale used in our core reference [1], which is last year’s ICLR paper**. It is also larger than the model sizes used in many recent studies on data filtering **[9,10,11,12,3]**. Therefore, our setup meets the standard level of rigor, and the results demonstrate that our method operates consistently across multiple scales. Also, the theoretical reason why our method yields performance gains regardless of scale is also well explained in **[W1]** and in response to **HKmF [W1]**.
>
> For this experiment, we rented four A100 GPUs (costing about \\$2k over two weeks, expected to total \\$4k by the end). We used a learning rate of 1.6e-4 following [1].
>
> $ \color{white}dd $
>
> **#. Request for more engagement in the discussion**
>
> We believe we have addressed all of the reviewer’s concerns. We would appreciate it if you could confirm whether there are any remaining issues, and if so, clearly indicate which points they refer to.

---

> > ### Author Response · Authors · 2025-12-03
> > **Additional experiment 2**
> >
> > #. Additional experiment on extended scale.
> >
> > EETable1. Overall average performance on benchmarks of the Qwen/Qwen2.5-3B model at each pre-training token duration (0.75B - 12B).
> >
> > | Token duration | 0.75B | 1.5B | 3B | 6B | 12B |
> > | --- | --- | --- | --- | --- | --- |
> > | Prior-based | 2.32 | 3.42 | 6.36 | 11.35 | 13.80 |
> > | PPL-based | 1.67 | 2.54 | 5.29 | 9.83 | 11.73 |
> >
> > EETable2. Results from training the Qwen/Qwen2.5-3B model for a 12B token duration (6B tokens × 2 repeat).
> >
> > |  | Avg | World knowledge | Commonsense reasoning | Language understanding | Symbolic problem solving | Reading comprehension |
> > | --- | --- | --- | --- | --- | --- | --- |
> > | Prior-based | 13.80 | 15.46 | 20.77 | 12.96 | 13.66 | 6.16 |
> > | PPL-based | 11.73 | 12.63 | 16.3 | 13.7 | 9.83 | 6.2 |
> >
> > Continuing from the last report, we present results from training a larger model (3B) at an expanded training scale (a total of 12B tokens, 6B tokens x 2 repeat), with core baselines (PPL- and prior-based). The superiority of our method remains consistent, demonstrating that the advantages of our approach hold even at larger scales.

---

### Author Response · Authors · 2025-11-29
**To ACs and future readers**

Dear ACs and future readers,

We sincerely appreciate your efforts and dedication.

We report that some of the concerns raised by reviewer **533u** and **edXW** appear to stem from not fully reading parts that are already sufficiently explained in the paper or from misunderstandings of our method. We notified them of this in the response and asked them to verify it again, but these reviewers did not participate in the discussion.

We provide the list of references below for your convenience.

**(1) Already explained in the paper:**

- 533u: [W1, W2, Q1]
      (W1 : §B.1, referred from §4, line 452/ W2Q1: §A, referred from §4.1)

- edXW: [W4, Q1, Q4, Q5]
(W4Q4Q5 : B.2 line 939, referred from  §4, line 452 )

**(2) Based on misunderstanding:**

- 533u: [W1]

- edXW : [W3, Q3, W2, Q4]

$ \color{white} $

The only remaining viable concern from these reviewers is that we have to experiment on a larger scale, such as pretraining 7B and 70B models on 100B token duration. Although their point has some validity, to the best of my knowledge, only a limited number of labs in academia have the capacity to perform such experiments, and the lab I belong to was no exception. We are concerned that such reviews may discourage interest and broad participation from the academic community, in this valuable topic of data filtering.

$ \color{white} $

Nevertheless, in the response, we provided the following explanation and experiments to further support our rigor (see response to **edXW [W1, and “additional experiments”] for detail**).

(1) We provided experiments at an extended scale and demonstrated that the trend remains consistent.

(2) We explained that previous representative data-filtering studies used model sizes comparable to ours. We also explained that our experimental scale is rigorous considering previous works, including out primary reference [1]. which is last year's ICLR paper.

(3) We provided an explanation of why the effects of noisy data filtering can be reliably measured even at smaller scales.

(4) We provided an explanation of why small-scale experiments can still indicate advantageous impacts on more complex tasks (e.g., MMLU-pro). We also provided an indirect evaluation supporting this.

(5) We provide a theoretical guarantee showing that our method necessarily exhibits noise-detecting behavior regardless of scale. (See response to **HKmF[W1]**)

$ \color{white} $

**#. For your convenience, we also provide brief references to frequently raised or important points and indicate where each is addressed.**

(1) Loss of rare but valuable data

: We have already addressed this issue very thoroughly in the paper **(in §B.1, referred from** §**4, line 452)**, and referred to it in the response (**8chv [W3]**).

(2) A request for a theoretical justification.

: We provided a more detailed explanation to **HKmF[W1].**

(3) Whether it works in other languages

: We provided additional explanations and experiments. (**8chv [W1]**)

(4) Whether it works in other tokenizers.

: This was already demonstrated experimentally in the paper (**§B.2, referred from §4.1, line 452)** , but we provided additional explanations. (**8chv [W1]**)

(5) Request for covering other baselines

: We provided additional explanations and experiments. **(HKmF[W2])**

$ \color{white} $

We will make sure that all the meaningful discussions raised here are reflected in the camera-ready version.

---

$ \color{white} $

[1] Z. Ankner et al., Perplexed by perplexity: Perplexity-based data pruning with small reference models, ICLR 2025

---

### Meta-Review · Area_Chair_FCGN · 2026-01-03

**Summary:**

This submission proposes a prior-based filtering criterion using corpus-level token frequency statistics (document-level mean/std of token priors) as a model-free, extremely efficient alternative to perplexity-based data pruning.

Reviewers’ main concerns centered on the following aspects:

(i) whether the observed gains are convincing at the reported experimental scale (mostly ≤1.5B models and ~6B–12B token durations) and whether conclusions would persist at 7B–70B / 100B-token regimes;

(ii) limited baseline breadth beyond PPL and DSIR (e.g., other lightweight heuristics/classifier-based filters), and sensitivity to hyperparameters such as filtering thresholds/ratios;

(iii) potential over-filtering of rare but high-value content (minority languages, niche domains, symbolic data), and the degree to which token-frequency proxies can capture semantic coherence compared to PPL;

(iv) clarity/positioning, with some reviewers viewing parts of the linguistic grounding and “learnability” claims as insufficiently quantified or potentially post-hoc.

In rebuttal, the authors addressed a substantial portion of these concerns by adding stronger comparisons, providing threshold/tokenizer ablations, and expanding scale via longer token-duration training and additional experiments on a larger 3B architecture (Qwen2.5-3B) where prior-based filtering remains consistently better than PPL across the reported benchmark suite. They also provided additional multilingual (e.g., Turkish) and rare-terminology robustness evidence, and clarified the intended scope/limitations of the method (structure/noise detection vs semantic nonsense where PPL may help). The primary remaining concern is the lack of direct validation at truly large model/data scales and the inherent limitation that frequency-based proxies may not fully capture semantic coherence; however, given the strong, reproducible empirical evidence across many tasks, the significant efficiency benefit, and the strengthened rebuttal, I consider the work a solid, practically impactful contribution with clear limitations to be stated in the final version.

**Reviewer Concerns:**

Reviewer 533u: Concerns about rare but valuable data, multilinguality, and baseline coverage were addressed through additional analyses and experiments, while the lack of direct validation at very large scales (7B–70B models, 100B+ tokens) remains outstanding.

Reviewer HKmF: Theoretical justification, robustness to mixed distributions, and baseline scope were clarified in the rebuttal, and no major concerns remain outstanding as the reviewer kept their score.

Reviewer edXW: Requests for larger-scale evaluation and sensitivity analyses were partially addressed with extended training and 3B-scale experiments, but experiments at extremely large data and model scales are still outstanding.

Reviewer 8chv: Multilingual robustness, tokenizer sensitivity, and preservation of rare data were addressed with additional evidence, while perceived limited novelty remains a subjective outstanding concern.

**Reviewer Scores:**

Reviewer 533u: may increase from 4 to 6

Reviewer edXW: w.h.p. 2 --> 6

Others remain unchanged.

---

### Decision · Program_Chairs · 2026-01-26

Accept (Poster)